bioinformatics/developmental biology/oceanography

atlantidae, ocean acidification, calcification, calcein indicator, gene expression, micro-CT

**Author for correspondence:**
Deborah Wall-Palmer
e-mail: dmwallpalmer@gmail.com

# The impacts of past, present and future ocean chemistry on predatory planktonic snails

Deborah Wall-Palmer[1], Lisette Mekkes[1,2],
Paula Ramos-Silva[1], Linda K. Dämmer[3,4],
Erica Goetze[5], Karel Bakker[3], Elza Duijm[1]
and Katja T. C. A. Peijnenburg[1,2]

[1]Plankton Diversity and Evolution, Naturalis Biodiversity Center, Leiden, The Netherlands
[2]Department of Freshwater and Marine Ecology, Institute for Biodiversity and Ecosystem Dynamics (IBED), University of Amsterdam, Amsterdam, The Netherlands
[3]Department of Ocean Systems, Royal Netherlands Institute for Sea Research (NIOZ), Texel, The Netherlands
[4]Environmental Geology, Department of Geology, Institute of Geosciences, University of Bonn, Bonn, Germany
[5]Department of Oceanography, University of Hawai'i at Mānoa, Honolulu, HI, USA

DW-P, 0000-0003-2356-6122; LM, 0000-0001-7208-1426;
PR-S, 0000-0002-0798-1724; LKD, 0000-0003-0891-9663;
EG, 0000-0002-7273-4359; KB, 0000-0002-8413-2398;
ED, 0000-0002-6383-3419; KTCAP, 0000-0001-7544-1079

The atlantid heteropods represent the only predatory, aragonite shelled zooplankton. Atlantid shell production is likely to be sensitive to ocean acidification (OA), and yet we know little about their mechanisms of calcification, or their response to changing ocean chemistry. Here, we present the first study into calcification and gene expression effects of short-term OA exposure on juvenile atlantids across three pH scenarios: mid-1960s, ambient and 2050 conditions. Calcification and gene expression indicate a distinct response to each treatment. Shell extension and shell volume were reduced from the mid-1960s to ambient conditions, suggesting that calcification is already limited in today's South Atlantic. However, shell extension increased from ambient to 2050 conditions. Genes involved in protein synthesis were consistently upregulated, whereas genes involved in organismal development were downregulated with decreasing pH. Biomineralization genes were upregulated in the mid-1960s and 2050 conditions, suggesting that any deviation from ambient carbonate chemistry causes stress, resulting in rapid shell growth. We conclude that atlantid calcification is likely to be negatively affected by future OA.

However, we also found that plentiful food increased shell extension and shell thickness, and so synergistic factors are likely to impact the resilience of atlantids in an acidifying ocean.

## 1. Introduction

In the marine realm, predators at all levels, from sharks to mesozooplankton, have a critical impact on the structure and function of ecosystems, and selective predators in particular are essential for maintaining biodiversity [1,2]. Consequently, the loss of predators can have a disproportionately large influence on ecosystem function [2]. The atlantid heteropods (Atlantidae, Pterotracheoidea) are one such family of selective predators in the plankton. These carnivorous holoplanktonic gastropods feed primarily on another key group of planktonic gastropods, the shelled pteropods (Thecosomata) [3,4]. Atlantids are sophisticated hunters, with complex eyes, a sucker on their fin to secure their prey and a flexible trunk-like proboscis to reach into shells and extract the prey within. Atlantids are also well adapted for their planktonic lifestyle, with a foot modified into a swimming fin and a small, thin aragonite shell (less than 14 mm diameter). Other heteropod families have reduced their shell with regards to body size (Carinariidae), or have lost it altogether in the adult stage (Pterotracheidae). The atlantids, however, are fully shelled because their shell is integral to their survival. Not only can they fully retract into the shell and seal themselves in with an operculum for protection, the adult atlantids also use their shell as a swimming appendage, which is paired with their single swimming fin, both being flapped in a coordinated wing-like manner to produce rapid directional propulsion [5,6]. The atlantid shell is already present upon hatching [3] and is likely important at the larval stage for physical protection, and also as ballast to allow rapid escape by sinking. A strong, well-constructed shell is therefore vital for the success of these ecologically important predators. However, almost nothing is known about the shell structure of atlantids and their mechanisms of calcification [7,8], despite likely being affected by imminent ocean changes, in particular rapid contemporary changes in ocean chemistry, and especially in the early life stages [9–12].

The current release of anthropogenic $CO_2$ into the atmosphere is causing a reduction in ocean pH at a rate unprecedented during the last 66 Myr [13–18]. The adverse consequences of this anthropogenic ocean acidification (OA) are being felt by many marine organisms [10,19]. Unlike the atlantids, their morphologically similar primary prey, the shelled pteropods, have been the focus of much calcification research over the last decade due to their sensitivity to changing ocean chemistry [20]. Recent research has confirmed negative effects of OA for shelled pteropods, including reduced calcification, increased shell dissolution and differential gene expression [20–24] and has highlighted them as useful OA-indicators, especially at higher latitudes [20,25,26]. The availability of food has also been found to influence pteropod calcification, with food deprivation acting as a synergistic stressor in OA experiments [27], and higher *in situ* food concentrations coinciding with thicker and larger pteropod shells in natural populations [28]. The atlantids, however, have been largely overlooked in OA research despite being similarly vulnerable.

The atlantid heteropods are thought to be among the most susceptible groups to OA and likely one of the first to experience it, due to a combination of three factors. First, the shell on which atlantids rely is constructed of aragonite [3,7]. Aragonite, a metastable form of calcium carbonate that is especially soluble in seawater [29], becomes difficult and energetically costly to produce under OA conditions [10,30] and aragonitic shells can dissolve if aragonite undersaturation occurs [21,31]. Second, atlantids inhabit the upper ocean, where the greatest proportion of anthropogenic $CO_2$ is being absorbed [12,32]. Atlantids also undergo diel vertical migrations over hundreds of meters [3,12]. With shoaling of the aragonite saturation horizon, they are increasingly likely to encounter deep waters that are undersaturated with respect to aragonite [33,34], thereby experiencing altered ocean chemistry across the vertical extent of their distributions. However, vertical migration could also help to increase the tolerance of atlantids to OA, as has been shown for some migrating pteropods and other zooplankton groups [35–37]. Third, atlantids can have high abundances in cold, mid-high latitude regions that have a higher capacity to absorb atmospheric $CO_2$, thus more rapidly becoming acidic compared to warmer regions [26,33]. Yet, despite the supposed vulnerability of aragonite shelled planktonic gastropods to OA, recent research suggests that atlantids and shelled pteropods survived past large-scale global change crises [38,39]. One of these events, the Paleocene-Eocene Thermal Maximum (PETM), is the most analogous geological event to the current Anthropogene climate crisis [40]. This suggests that atlantids and pteropods may be more resilient to OA than expected.

Although superficially similar, shelled pteropods and atlantid heteropods are phylogenetically and ecologically very distinct, having evolutionarily independent origins and occupying different trophic

levels. These differences could mean that they have very different physiological responses to changing ocean carbonate chemistry. While shelled pteropods are particle-feeders via mucous webs, juvenile atlantids feed on algae using a ciliated velum and adult atlantids are selective, visual predators. As such, atlantids are the only aragonite shelled predatory plankton, and are uniquely positioned to indicate the effects of changing ocean chemistry on higher trophic levels. Atlantids are found in the epipelagic zone of open waters mainly from tropical to temperate latitudes, although there are two cold water species [41]. *Atlanta ariejansseni* is the most southerly distributed atlantid species, with a circumglobal distribution restricted to the Southern Subtropical Convergence Zone (SSTC) between 35–48°S, where this species can reach abundances of up to approximately 200 individuals per 1000 m$^3$ and likely represents an important predator and calcifier within the plankton [42]. The SSTC is a narrow region at the boundary between warmer, more saline subtropical waters to the north, and the colder, fresher Sub-Antarctic Zone to the south [43]. As such, the SSTC is a highly variable region with strong gradients in salinity and temperature, and is expected to experience considerable ocean change, in particular OA [42,44,45]. The cold water distribution and relatively high abundance of *A. ariejansseni* make it an excellent candidate as an OA sentinel species. In addition, juvenile atlantids are relatively easily maintained under laboratory conditions because they feed on algae using their ciliated velum.

Here, we present results of the first growth and OA experiments focussed on heteropods, and the first transcriptome of a heteropod. We use a thorough multi-disciplinary approach, combining fluorescence microscopy ($n = 184$) and micro-CT scanning ($n = 43$) of the same individuals to quantify shell growth, as well as RNA sequencing of individuals from the same experiments to detect responses at the molecular level ($n = 6$ samples of 8–10 pooled individuals) to address the following questions: (1) What is the rate of atlantid shell growth under current ambient conditions in the South Atlantic Ocean? (2) Has atlantid calcification already altered in response to recent changes in high latitude ocean carbonate chemistry? (3) Will future ocean conditions lead to a decline in atlantid calcification, similar to the response found for shelled pteropods? This study demonstrates the suitability of juvenile atlantids as OA sentinels, providing important insight into atlantid biomineralization through morphological measurements and transcriptome analyses. As ecologically important predators within the plankton, understanding the likely effects of ocean changes on atlantid calcification will also help in understanding the impacts of OA on the pelagic ecosystem as a whole.

## 2. Methods

### 2.1. Specimen collection and staining

Although the ecological importance of atlantids lies mainly in their adult predatory stage, we chose to use specimens in the herbivorous juvenile stage for our experiments. This decision was made for a number of reasons. Most importantly, early life stages of molluscs are known to be particularly sensitive to OA, exhibiting a reduction in development and metabolism with increasingly acidic conditions [9–11]. We aimed to capture this response to OA at a critical life stage. In addition, it is certain that juvenile specimens are not fully grown and would have the opportunity to calcify within the experiments, which cannot be guaranteed for adult specimens. Finally, with the limitations of maintaining pteropods in laboratory conditions [46], securing a constant food source for adult atlantids was problematic. The herbivorous juvenile stages could easily be provided with an algal food source at the onset of the experiments without need to disturb the experiments once they were running. Juvenile atlantids likely have periods of active feeding and resting [41]. A replete food supply was added to be sufficient for several days under the possibility that the animals feed near continuously under experimental conditions.

Specimens of *Atlanta ariejansseni* were collected in the Southern Subtropical Convergence Zone during the Atlantic Meridional Transect (AMT) 27 (DY084/085) cruise of the *RRS Discovery*. No permissions/permits were required to carry out this fieldwork. Animals for growth experiments were collected on the 24th October 2017 at 35°58 S, 27°57 W and for OA experiments on the 26th October 2017 at 41°09 S, 30°00 W. For both experiments, samples were collected using a 1 m diameter ring net with 200 µm mesh and a closed cod-end for three slow, short (20 min) oblique tows to a maximum depth of 100 m. Samples were collected during hours of darkness between 00 : 38 and 01 : 57. Specimens of *A. ariejansseni* were immediately sorted from the net samples using a light microscope and placed in calcein indicator for 2 h in the dark (MERCK Calcein indicator for metal determination, CAS 1461-15-0, concentration 50 mg l$^{-1}$ in seawater filtered through a 0.2 µm filter). Specimens were then gently rinsed with 0.2 µm filtered seawater and introduced into the experimental carboys.

## 2.2. OA experiment

To evaluate the effects of ocean carbonate chemistry on atlantid calcification, specimens of *A. ariejansseni* were incubated for 3 days across realistic recent-past, ambient and near-future pH scenarios. We applied a past scenario of 0.05 pH units higher than ambient (ambient pH 8.14 ± 0.02, past pH 8.19 ± 0.02, table 1) that is approximately equivalent to the mid-1960s (assuming a decrease in pH of 0.001 units per year in this region) [18], and a future OA scenario of 0.11 pH units lower than ambient (pH 8.03 ± 0.00, table 1), which is approximately equivalent to expectations for the year 2050 in the South Atlantic Ocean (under IPCC Representative Concentration Pathway RCP8.5) [13,47]. Aragonite saturation was maintained in all scenarios ($\Omega > 1.82$).

Three replicate experiments were conducted for each of the three treatments ($n = 9$, plus three controls, total $n = 12$ carboys). To ensure consistency of water chemistry across the replicates of each treatment, water was pre-treated in barrels before being used to fill experiment carboys. Surface seawater (from approx. 10 m water depth) was filtered at 0.2 µm into four 60 l barrels, which underwent the following treatments. In one barrel, lowered pH was achieved by bubbling 795 ppm $CO_2$ in air through the water for 12 h. In a second barrel, higher pH was achieved by bubbling 180 ppm $CO_2$ in air through the water for 12 h. The final two barrels of ambient and control (ambient) water were not subjected to any gas bubbling. During gas bubbling, all water was maintained at ambient ocean temperatures (at the depth of specimen collection 0–100 m), between 14 and 16°C within a temperature controlled room on board the *RRS Discovery*. Temperature, salinity and pH (HANNA HI5522-02) were measured from the four barrels after 12 h, and samples to measure Dissolved Inorganic Carbon (DIC) concentration were collected. Immediately prior to specimen collection, three carboys of six litres were filled for each of the treated and ambient seawaters. Three further carboys were filled with ambient seawater to act as controls (no specimens added; water chemistry analyses were successful for two control carboys). *In situ* phytoplankton concentrations at the study sites over the depth sampled for specimens (0–100 m) was 0.16–0.42 µg l$^{-1}$ [48]. To ensure that food was not limiting calcification rates, freeze dried algae (a mixture of 33.3% *Phaeodactylum*, 33.3% *Nannochloropsis*, 33.3% *Tetraselmis*) was added to each of the carboys (including the controls) at a concentration of 0.6 mg l$^{-1}$ (0.2 mg l day$^{-1}$; 3.6 mg per carboy). Algae was only added at the beginning of the experiment to prevent disturbance of the specimens and to avoid exposing the experimental seawater to the atmosphere. A high concentration (approx. 500–1000 times more than the *in situ* concentration) of algae was therefore added to ensure that *ad libitum* feeding was possible for the duration of the experiment.

Calcein-stained specimens ($n = 274$, 217 for physical measurements, 57 for gene expression analysis) were introduced into the carboys in an arbitrary order. Between 90 and 92 juveniles were exposed to each of the three treatments. Specimens of *A. ariejansseni* were identified by their shell morphology [42], and juveniles were recognized based on size, the presence of the velum and the absence of black eye pigmentation. Carboys were sealed air-tight with no head space and immediately incubated at ambient temperature within a temperature controlled room (14–16°C). Blackout fabric was draped over the carboys to maintain low light levels. The carboys were incubated for 3 days. At the end of the third day, temperature, salinity and pH were measured, and DIC concentration samples were collected from all carboys. Specimens were removed from the carboys, and examined under a light microscope to verify that they were still alive (movement). Twenty live juveniles were pooled for each treatment (from one replicate), preserved in RNAlater (Invitrogen) and frozen for RNAseq analyses. The remaining specimens were flash frozen in liquid nitrogen and stored at −20°C until analysis.

## 2.3. Growth rate experiment

Surface seawater was filtered at 0.2 µm and maintained at ambient temperature in a 6 l carboy. Temperature, salinity and pH were measured and samples for DIC concentration were collected prior to adding specimens. Freeze dried algae were added to the carboy at a concentration of 4.8 mg l$^{-1}$, to allow for approximately 24 days of feeding (0.2 mg l day$^{-1}$. 28.8 mg per carboy). Calcein-stained juvenile specimens ($n = 57$) were introduced into the carboy, which was immediately sealed and incubated at ambient temperature, covered with blackout fabric. After 3 days, temperature, salinity and pH measurements were made, and a sample for DIC concentration was collected. Up to ten live individuals were removed and flash frozen in liquid nitrogen and stored at −20°C until analysis. Any dead specimens retrieved at this stage were removed from the experiment and discarded (total for the whole experiment $n = 13$). Subsequent to sampling, the carboy was sealed and returned to ambient,

**Table 1.** Measured and calculated (using CO2SYS) carbonate system parameters for the three ocean acidification treatments and the resulting shell growth of *Atlanta ariejansseni* (averaged across all replicates). $N1$ is the number of specimens measured for the shell extension (total $n = 184$) and $N2$ is the number of specimens measured for volume, thickness and diameter ($n = 43$). Values are presented $\pm 1$ s.d. when averaged over replicates.

| sample | day | mean measured DIC (μmol k⁻¹g) | mean calculated TA (μmol k⁻¹g) | mean measured pH (NBS) | mean calculated $pCO_2$ | mean calculated ΩAr. | N1 | mean shell extension (μm) | N2 | mean volume of shell grown (1000 μm³) | mean thickness of shell pre-exp (μm) | mean thickness of shell grown (μm) | mean maximum shell diameter (μm) |
|---|---|---|---|---|---|---|---|---|---|---|---|---|---|
| ambient control | 0 | 2112 | 2338 | 8.16 | 410 | 2.51 | — | — | — | — | — | — | — |
| ambient control | 3 | 2103 ± 2 | 2308 ± 0 | 8.13 ± 0.00 | 441 ± 9 | 2.27 ± 0.02 | — | — | — | — | — | — | — |
| mid-1960s | 0 | 2110 | 2352 | 8.18 | 391 | 2.65 | — | — | — | — | — | — | — |
| mid-1960s | 3 | 2109 ± 6 | 2341 ± 5 | 8.19 ± 0.02 | 378 ± 18 | 2.53 ± 0.10 | 61 | 136 ± 22 | 14 | 963 ± 210 | 5.81 ± 0.89 | 11.58 ± 1.81 | 400 ± 49 |
| ambient | 0 | 2110 | 2339 | 8.16 | 405 | 2.53 | — | — | — | — | — | — | — |
| ambient | 3 | 2139 ± 10 | 2350 ± 2 | 8.14 ± 0.02 | 425 ± 20 | 2.33 ± 0.08 | 63 | 124 ± 16 | 15 | 755 ± 246 | 5.96 ± 0.61 | 11.51 ± 1.30 | 361 ± 72 |
| 2050 | 0 | 2150 | 2329 | 8.05 | 553 | 2.05 | — | — | — | — | — | — | — |
| 2050 | 3 | 2154 ± 13 | 2312 ± 11 | 8.03 ± 0.00 | 564 ± 5 | 1.84 ± 0.02 | 60 | 155 ± 26 | 14 | 871 ± 270 | 5.68 ± 0.73 | 10.58 ± 1.29 | 386 ± 96 |

**Table 2.** Typical shell growth of juvenile *Atlanta ariejansseni* at ambient conditions over 11 days. $N$ is the number of specimens sampled on each day.

| sampling day | N | measured DIC (μmol kg⁻¹) | calculated TA (μmol kg⁻¹) | measured pH (NBS) | calculated $pCO_2$ | calculated ΩAr | mean total shell extension (μm) ± 1SD | maximum total shell extension (μm) | mean shell extension per day (μm) | maximum shell extension per day (μm) |
|---|---|---|---|---|---|---|---|---|---|---|
| 0 | — | 2112 | 2358 | 8.15 | 426 | 2.73 | — | — | — | — |
| 3 | 8 | 2130 | 2345 | 8.15 | 417 | 2.37 | 208 ± 65 | 299 | 69 | 100 |
| 5 | 10 | 2141 | 2368 | 8.17 | 399 | 2.49 | 231 ± 77 | 374 | 46 | 75 |
| 7 | 10 | 2135 | 2360 | 8.17 | 394 | 2.45 | 306 ± 87 | 456 | 44 | 65 |
| 9 | 9 | 2151 | 2364 | 8.14 | 426 | 2.35 | 271 ± 109 | 486 | 30 | 54 |
| 11 | 7 | 2173 | 2378 | 8.14 | 437 | 2.29 | 408 ± 148 | 599 | 37 | 54 |

dark conditions. Sampling was carried out in the same way for water parameters and specimens approximately every 2 days. The experiment was terminated at 11 days because some specimens metamorphosed and began to cannibalize other animals in the experiment.

## 2.4. Water chemistry (from barrels and carboys)

Dissolved inorganic carbon (DIC) samples were filtered into 5 ml glass vials. The water samples contained no head space and were poisoned with 15 µl of saturated mercury (II) chloride ($HgCl_2$). Analysis of DIC was carried out at the Royal Netherlands Institute for Sea Research (NIOZ), Texel, The Netherlands, using a Technicon Traacs 800 autoanalyzer spectrophotometric system following the methodology of Stoll *et al.* [49]. pH was measured on the NBS scale using a research grade benchtop pH metre (HANNA HI5522-02, accuracy ±0.002, readable to 0.1 mV and 0.001 pH as recommended for OA research [50]) and a glass electrode. The pH metre was regularly calibrated using NBS standards (HANNA Millesimal Buffer Range, accuracy ±0.002 pH). Other carbonate system parameters were calculated from the measured DIC and measured pH [51] using CO2SYS (Excel V2.3) [52]. The calculation used the constants K1 and K2 from Mehrbach *et al.* [53] refitted by Dickson & Millero [54] and the KHSO4 dissociation constant of Dickson [55]. Nutrient concentrations (P and Si) were measured from surface CTD samples in the regions where water was collected for the experiments [56].

## 2.5. Shell extension

One method for quantifying calcification is to measure the length of the piece of shell that grew during the experiment e.g. [57], herein described as the shell extension. Shell cleaning and fluorescent imaging to measure this shell extension was carried out at the Royal Netherlands Institute for Sea Research (NIOZ), Texel, The Netherlands. Organic material was removed from the shells by oxidizing the specimens in a Tracerlab low temperature (approx. 100°C) asher. This ensured minimal damage compared to chemical and physical washing techniques. Specimens were air dried for 24 h and then oxidized in the low temperature asher for 5 h. Specimens were then gently rinsed with ethanol and ultra-high purity water (MilliQ) to remove any ash residue, and dried in a cool oven (40°C) for 15 min. Specimens were imaged using a Zeiss Axioplan 2 microscope with a Colibri light source and filter (excitation 485/20, FT 510, emission 515–565) producing a final wavelength of 515–565 nm. The extent of fluorescent shell was measured along the suture between the last whorl and the preceding whorl using the software FIJI (ImageJ) [58]. Despite extreme care being taken during shell handling, the growing edge of some shells was damaged, providing only a minimum measure of shell extension. Therefore, for the OA experiments, severely damaged specimens ($n = 31$), where the shell edge was broken back to the calcein stained region (start of the experiment), were not included in subsequent analyses.

## 2.6. Shell thickness and volume

Specimens were visually inspected using light microscopy to determine whether there was any damage at the growing edge of the shell. Between 14 and 15 undamaged specimens were selected from each treatment (total $n = 43$ specimens) and scanned using a Zeiss Xradia 520 Versa microCT at Naturalis Biodiversity Center, Leiden, The Netherlands. Scans were carried out using between 140/10 and 150/10 kV $W^{-1}$ for between 2 and 3 h per specimen. The scan resolution was 0.54–0.68 µm with an exposure time of 5–10 s. Data were processed using the software Avizo 2019.1 (Thermo Fisher). Shells were segmented to separate the part of the shell that had grown during the experiment. This was achieved by manually matching the fluorescent images to the microCT thickness map using whorl counting and other landmarks such as growth lines and repair marks on the shells (figure 1*a–h*). The segmented shells were then analysed for the volume and mean thickness of the shell grown during the experiment. MicroCT images were also measured to determine maximum shell diameter.

## 2.7. Statistical analyses

To determine correlations between variables, for example between shell thickness and shell diameter, Pearson's correlation coefficient was used, and to confirm correlations between the carbonate system parameters, a full pairwise matrix of Pearson's correlation coefficients was made. Differences between the growth measurement data for the OA experiments (shell extension, shell thickness and shell

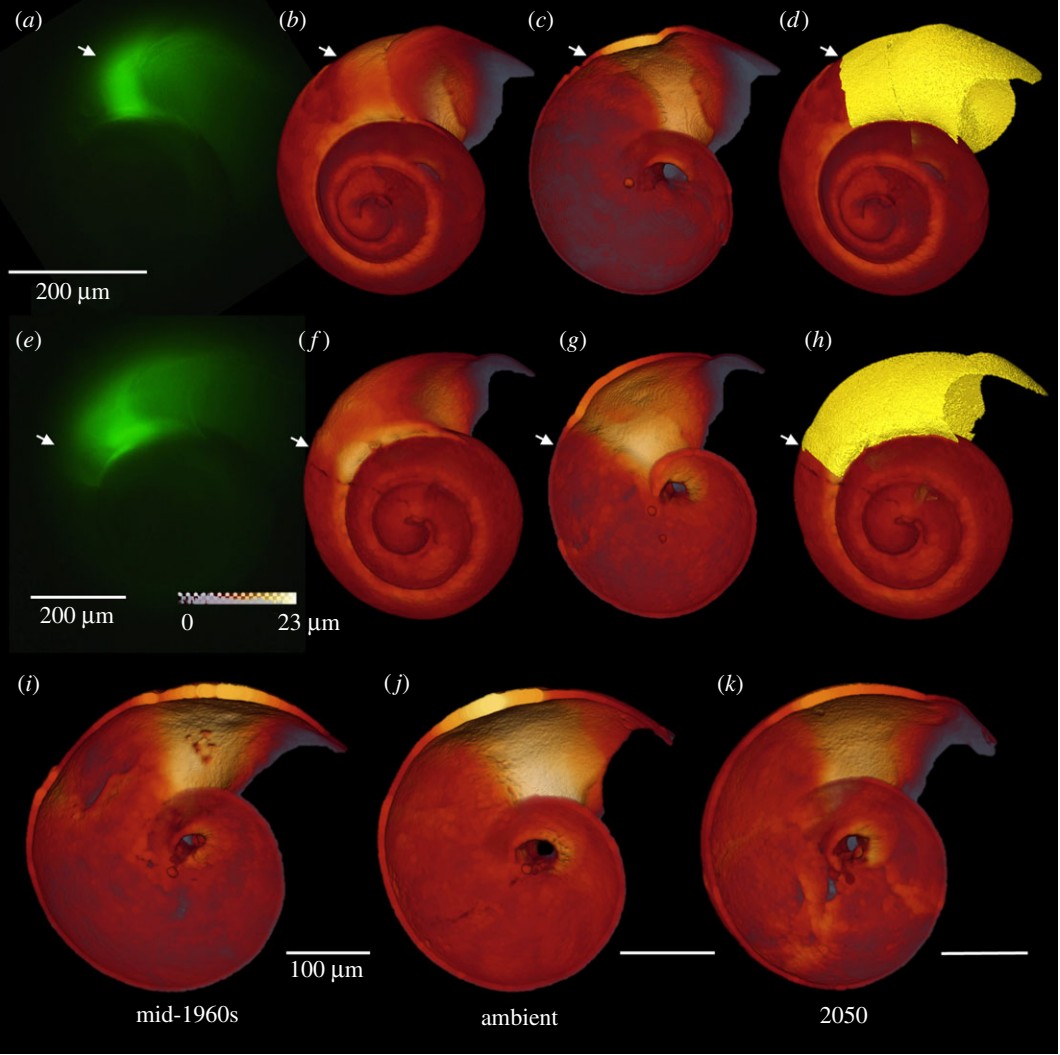

**Figure 1.** Quantifying calcification of juvenile *Atlanta ariejansseni*. (*a,e*) Shell extension of the shell was measured using the fluorescence images. (*b,c,f,g*) The position of the onset of the experiment/glow was then identified in thickness maps produced using micro-CT. White arrows show the onset of the experiment. Thickness maps presented in the 'glow' colour scheme. (*d,h*) The shell was segmented to isolate the part of the shell grown during the experiment. (*a–d,i*) mid-1960s, (*e–h,j*) ambient, (*k*) future 2050.

volume), and shell extension per sampling day for the ambient growth experiment were tested using a one-way PERMANOVA, including pairwise comparisons to detect more detailed differences between treatments. Comparisons of shell thickness (shell grown during the OA experiment versus shell grown *in situ*), and comparisons of shell extension resulting from different food concentrations were made using a *t*-test. Statistical analyses were carried out using PAST v. 4.04 [59]. For all statistical analyses, *p*-values < 0.05 were considered to be significant.

## 2.8. RNA extraction and sequencing

Due to the small size of *Atlanta ariejansseni* juveniles (mean diameter of a subset 381 ± 73 µm, $n = 39$), total RNA was extracted from two samples of 8–10 individuals pooled from each treatment using the RNeasy Plus Micro Kit (QIAGEN). This sampling provided two replicates per treatment (six samples in total) and was chosen to ensure that there was enough RNA for downstream processing. Each sample of total RNA was analysed for quantity and quality using the Bioanalyzer 2100 (Agilent Technologies) with an RNA 6000 Nano Chip. All samples had RIN scores ranging from 7.0 to 9.4 and were used for library preparation and sequencing. Libraries ($n = 6$) were generated with the NEBNext® Ultra II Directional Library Prep Kit for Illumina (New England BioLabs) using the manufacturer's protocol

for Poly(A) mRNA magnetic isolation from 1 µg total RNA per sample. Total RNA was added to NEBNext Sample Purification beads to isolate the mRNA. Purified mRNA was then fragmented into approx. 300 base-pair (bp) fragments and reverse transcribed into cDNA using dUTPs in the synthesis of the second strand. cDNA fragments were size selected and amplified with 8–9 PCR cycles using NEBNEXT Multiplex Dual Index kit (New England BioLabs) according to the manufacturer's instructions. Libraries were checked for quantity and quality on the Bioanalyzer 2100 using an Agilent DNA High Sensitivity Chip. Average library sizes of 420 up to 450 bps (approx. 300 bp insert +128 bp sequencing adapters) were accessed using the Agilent Bioanalyzer 2100. Sequencing was performed at BaseClear BV Leiden on an Illumina NovaSeq 6000 platform using paired-end 150 base-pair sequences. All six libraries were sequenced producing a minimum of 6 giga base pairs (Gb) per library.

## 2.9. *De novo* assembly and data analysis

Raw reads were processed using trimmomatic (v. 0.38 [60]) to remove adapter sequences and reads lower than 36 bps, and checked for quality using FastQC (v. 0.11.8). Trimmed reads were pooled and assembled with Trinity v. 2.8.4 [61] using default parameters. Open reading frames (ORFs) of the *de novo* transcriptome assembly were predicted using Transdecoder v. 5.5.0 [61]. The ORFs of the longest isoforms from each trinity locus were blasted against a subset of the NCBI nr database (release from 9/20/19) including all Mollusca (txid6447), Stramenophiles (txid33634) and Viridiplantae (txid33090). Contigs having a best hit with molluscan sequences (e-value $< 10 \times 10^{-5}$) were considered *bona fide Atlanta ariejansseni* transcripts; all the other contigs (for example, derived from the mixture of algae fed to the animals or other potential contaminants) were removed from the assembly. After this filtering, the distribution of GC content in the assembly appeared unimodal (electronic supplementary material, figure S1) suggesting the major sources of contamination were removed without compromising the transcriptome completeness as determined with BUSCO [62] (electronic supplementary material, table S1). Next, transcript quantifications were estimated based on the raw reads and the raw transcriptome assembly as reference, using Salmon [63]. Only quantifications of transcripts present in the clean transcriptome assembly were used in differential gene expression estimation using the DESeq2 package [64] in pairwise comparisons between the ambient versus higher mid-1960s pH and lower 2050 versus ambient pH. Significant differentially expressed genes were selected based on P-adj values $< 0.05$ corrected for multiple testing with the Benjamini–Hochberg procedure, which controls false discovery rate (FDR) (electronic supplementary material, tables S2 and S3). Annotation of the clean transcriptome assembly was performed using the Trinotate v. 3.2.0 pipeline, which levered the results from different functional annotation strategies including homology searches using BLAST+ against Swissprot (release October 2019) and protein domain detection using HMMER [65] against PFAM [66] (release September 2018). Gene ontology (GO) terms obtained from this annotation strategy (electronic supplementary material, table S4) were trimmed using the GOSlimmer tool [67] followed by enrichment analyses using the GO-MWU method described in [68]. This method used adaptive clustering of GO categories and Mann–Whitney $U$ tests based on ranking of signed log $p$-values to identify over-represented GO terms in the categories 'Biological Process' and 'Molecular Function'. In addition, genes were grouped in 10 main categories (i.e. putative processes or other) according to their BLAST+ best hits in RefSeq or Swissprot (releases October 2019) and associated GO terms (electronic supplementary material, tables S2 and S3): 'immune response', 'protein synthesis', 'protein degradation', 'biomineralization', 'carbohydrate metabolism', 'development/morphogenesis', 'ion transport', 'oxidation-reduction', 'lipid metabolism' and 'other'. Gene expression heatmaps with hierarchical clustering of expression profiles were created with ClustVis [69].

# 3. Results

## 3.1. Water chemistry

Across all treatments, the pH of the experimental carboys was stable from the start to the end of the experiment and remained fairly consistent between replicates (table 1). The pH of two control carboys containing ambient water and no specimens also remained stable over the 3 days and did not differ from the ambient experiment. DIC analyses for the third control carboy failed, therefore results are only reported for the two control carboys with a full suite of water chemistry data. The water remained supersaturated with regards to aragonite throughout all treatments (table 1) and no signs of

**Table 3.** Results of one-way PERMANOVA analyses ($p$ and $F$ values) carried out on the shell extension measurements of *A. ariejansseni* per sampling day for the ambient growth experiment.

| | sampling day | 3 | 5 | 7 | 9 |
|---|---|---|---|---|---|
| pairwise *p*-values | 5 | 0.5140 | | | |
| | 7 | 0.0193[a] | 0.0587 | | |
| | 9 | 0.1802 | 0.3736 | 0.4444 | |
| | 11 | 0.0028[a] | 0.0054[a] | 0.0941 | 0.0522 |
| | F | 4.825 | | | |
| | p | 0.0028 | | | |

[a]Denotes significant *p*-values (less than 0.05).

shell dissolution were observed (surface etching or clouding of the shells). All carbonate system parameters are correlated (Pearson $r = -0.996$–$0.977$, $p = <0.002$), except DIC and total alkalinity (Pearson $r = -0.114$, $p = 0.687$).

## 3.2. Shell growth under ambient ocean chemistry conditions

The calcein indicator, which is incorporated into the shells during growth and can be detected using fluorescence microscopy, was only integrated into the apertural/growing edge of the shell, suggesting that the shell is not thickened from inside as is observed in some pteropods [24,70]. Shell extension is therefore an informative measure of shell growth. Some small repairs from the inside surface of the shell, similar to those found in pteropods [71] were also observed (electronic supplementary material, figure S2). Towards the end of the ambient shell growth experiment (days 9 and 11), several individuals were observed to have undergone metamorphosis, having lost their velum and developed their swimming fin.

In the ambient growth experiment, specimens grew up to 99 µm (shell extension) per day ($n = 44$, table 2). Individuals grew significantly during the experiment (PERMANOVA, $F = 4.825$, $p = 0.003$). However, growth between sampling days was not always significant (table 3) and there was a noticeable drop in mean shell extension at day 9 (table 2 and figure 2a). Mean shell extension (all specimens) and maximum shell extension (largest specimen for each collection day) varied from 30–69 µm and 54–99 µm per day, respectively, and both were found to decrease exponentially with age (figure 2b). This pattern may be due to the relatively broader surface of the shell as the shell shape inflates with increasing age and number of shell whorls, such that the amount of shell produced may be approximately the same. Adult specimens of *A. ariejansseni* were found to exhibit a comparatively low growth rate of approximately 25 µm per day (mean of 5 adult specimens). Assuming that the shell of *A. ariejansseni* follows the exponential decrease in shell extension identified by the mean shell extension per day (ln $-26.89 \times$ days $+ 95.076$), and assuming a minimum growth rate of 25 µm per day, it would take around 116 days for an *A. ariejansseni* specimen to grow from hatching to full adult size (approx. 3200 µm of shell extension measured along whorl suture).

## 3.3. The effects of OA on calcification

Calcification was measured in three ways: mean shell extension was measured from fluorescence images, while the volume of shell grown during the experiment (referred to as shell volume) and the mean thickness of the shell grown during the experiment (referred to as shell thickness) was quantified using micro-CT scanning (figures 1 and 3). Mortality was extremely low across all treatments, with only a single specimen (ambient treatment) having died during the experiment.

Individuals grew significantly less shell material under the ambient conditions compared to the mid-1960s treatment (figure 3a, shorter shell extension PERMANOVA pairwise comparisons, $p = <0.001$; and lower shell volume $p = 0.021$, table 4). Shell thickness remained similar between the mid-1960s and ambient treatments (figure 3c, $p = 0.895$).

A further reduction in calcification with decreasing pH was not observed from the ambient treatment to the 2050 treatment. Individuals grown under 2050 conditions produced the same volume of shell as individuals grown under ambient conditions (figure 3b, PERMANOVA pairwise comparisons, $p =$

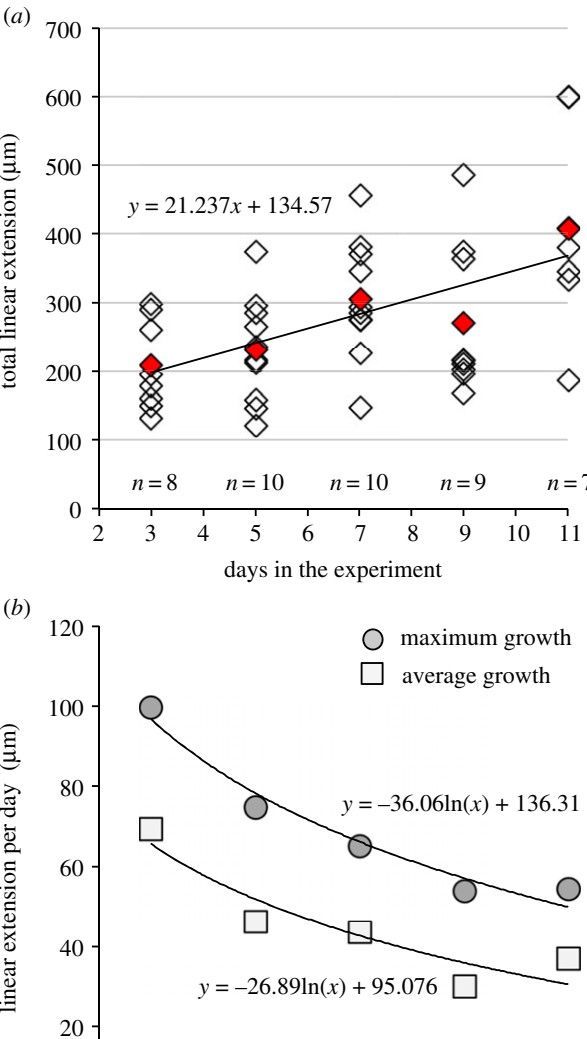

**Figure 2.** Shell extension of *Atlanta ariejansseni* juveniles in ambient conditions over 11 days. (*a*) Total shell extension for each individual. Red diamonds represent mean total shell extension for each sampling day. Line calculated using all data points. (*b*) Maximum and mean shell extension per day for each sampling day decreases exponentially and can be used to estimate the time taken to grown to full adult size, assuming constant conditions.

0.247, table 4), however, shell extension was significantly greater under the 2050 treatment than under ambient conditions (figure 3*a*, $p = <0.001$). Although there was no significant difference in shell thickness (figure 3*c*, $p = 0.071$), shells grown under the 2050 treatment were found to be generally thinner (10.6 µm ± 1.3, table 1) than those grown under both ambient (11.6 µm ± 1.8) and mid-1960s (11.5 µm ± 1.3) conditions.

Aside from the differences between treatments detailed above, the mean thickness of the shell grown during the OA experiments was significantly higher (1.6 to 2.6 thicker) than the shell grown *in situ* prior to the experiment for all shells measured in all treatments (*t*-test, $t = -20.720$, $p = <0.001$, $n = 41$). All micro-CT scanned individuals show an initial thickening of the shell ($n = 43$, figure 1*c*,*g*,*i*–*k*) that coincides with the onset of the experiment (apart from two specimens in which the thickening occurred after the onset).

Mean shell thickness was found to negatively correlate to shell diameter (Pearson $r = -0.669$, $p \leq 0.001$, $n = 39$), but only for shell grown prior to the experiments (electronic supplementary material, figure S4). The mean thickness of shell grown during the experiments was not correlated to shell diameter for any of the treatments.

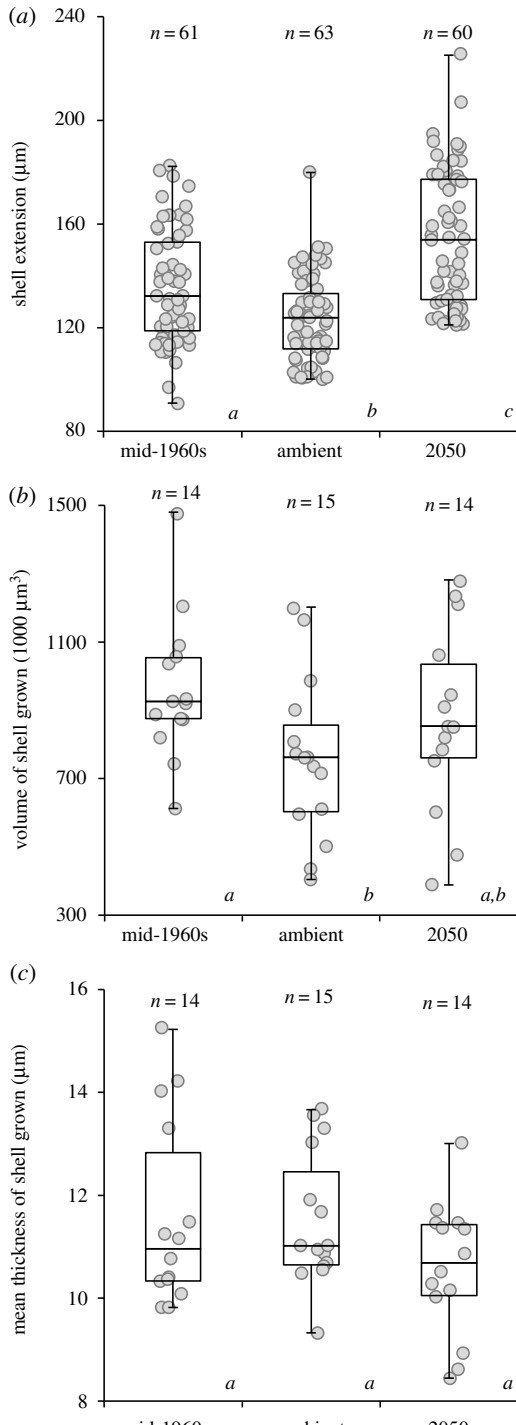

**Figure 3.** Shell growth of juvenile *Atlanta ariejansseni* was lower in ambient conditions when compared to mid-1960s conditions, but from ambient to 2050 conditions, longer shell was produced. (*a*) Shell extension data gathered across the three ocean chemistry conditions for 184 specimens (outliers were removed, see methods). (*b*) Shell volume and (*c*) Shell mean thickness grown under the three ocean chemistry conditions for a subset of 43 specimens. For the boxplots, horizontal lines are median values, boxes are 1st and 3rd quartiles, and bars show the minimum and maximum measurements. Scattered points show all measurements. Significant differences between treatments are denoted by italic letters below each box.

## 3.4. The effects of differing food concentrations

Specimens sampled from the ambient growth experiment at day 3 ($n = 8$, pH 8.15) were compared to specimens from the ambient OA experiments (also incubated for 3 days, $n = 63$, pH 8.14 ± 0.02) to detect any effects of differing food concentrations on shell extension. A significant difference in the shell extension was found between the two groups of specimens (*t*-test, $t = 8.714$, $p = <0.001$).

**Table 4.** Results of one-way PERMANOVA analyses ($p$ and $F$ values) carried out on measurements of shell grown by juvenile *A. ariejansseni* during the OA experiments.

|  |  | shell extension | shell thickness | shell volume |
|---|---|---|---|---|
| pairwise *p*-values | mid-1960s versus ambient | 0.0006[a] | 0.8949 | 0.0207[a] |
|  | ambient versus 2050 | 0.0001[a] | 0.0702 | 0.2469 |
|  | mid-1960s versus 2050 | 0.0001[a] | 0.1077 | 0.3212 |
|  | *F* | 31.93 | 1.984 | 2.65 |
|  | *p* | 0.0001[a] | 0.1480 | 0.0874 |

[a]Denotes significant *p*-values (less than 0.05).

Specimens in the ambient growth experiment, which were given eight times more food, grew on average 1.62 times longer shell than those in the ambient OA experiment with the lower food concentration (mean shell extension 208 µm and 129 µm, respectively).

## 3.5. The effects of OA on gene expression

A significant response to the treatments was also observed at the gene expression level. Reads from each treatment were mapped to a nearly complete *A. ariejansseni de novo* reference transcriptome (electronic supplementary material, table S1) to obtain transcript and predicted gene counts. However, because we only have two biological replicates, our differential gene expression estimations are interpreted cautiously. Comparing the mid-1960s to the ambient conditions, 110 genes were differentially expressed (DE) (66 down and 44 upregulated), while from the ambient to the 2050 conditions there were 49 DE genes (12 down and 37 upregulated), with only 9 of them shared between treatments (electronic supplementary material, figure S3, adjusted $p$-value < 0.05). In total the DE genes account for approximately 0.5% of the *A. ariejansseni* transcriptome which is within the size range of previous transcriptomic responses to high $CO_2$ in pteropods (0.001 to 2.6%) [22,23,72,73] and copepods (0.25%) [74].

GO enrichment analysis was implemented to identify which molecular functions (MF) and biological processes (BP) were being repressed or activated in response to ocean carbonate chemistry. Genes with GO categories associated with protein synthesis were consistently upregulated with decreasing pH (figure 4a,b, in red), while genes with GO categories associated with morphogenesis and organismal development were downregulated (figure 4a,b, in blue). Similar DE GO categories have also been documented in OA studies with pteropods, though the direction of the response was found to be different, with most genes involved in protein synthesis being repressed in response to high $CO_2$ [75].

Significant transcriptional changes were further elucidated by the annotation of each DE gene. Based on the transcriptome annotation with BLAST (electronic supplementary material, tables S2–S4), most genes responsive to changes in the carbonate chemistry were potentially involved in nine functional categories, including (1) immune response, (2) protein synthesis, (3) protein degradation, (4) biomineralization, (5) carbohydrate metabolism, (6) morphogenesis, development and nervous system, (7) ion transport, (8) oxidative and (9) lipid metabolism (electronic supplementary material, tables S2 and S3). A heatmap comparing the transcript levels of genes involved in protein synthesis (figure 4d) strongly corroborates the results obtained from the GO enrichment analysis, with most genes being upregulated from mid-1960s to ambient and from ambient to 2050 conditions, i.e. in the direction of decreasing pH. A more complex pattern was observed for genes involved in morphogenesis, development and the nervous system (figure 4c). Several genes were strongly upregulated in the mid-1960s, and three genes were strongly upregulated in the 2050 treatment.

Expression patterns of genes involved in oxidative metabolism suggest that both the mid-1960s and 2050 treatments had an effect on respiration (figure 5a). In particular, two genes associated with mitochondria, were strongly upregulated in the future treatment. Consistent patterns of upregulation of mitochondrial genes were also found for the pteropod *Limacina retroversa* at low pH [22] but contrast with the downregulation of these genes in the pteropod *Heliconoides inflatus* [23]. In turn, genes coding for heme-binding proteins (involved in oxygen transport) were equally upregulated in mid-1960s and 2050 conditions while enzymes with oxidoreductase activity were downregulated (figure 5a).

A significant number of genes potentially involved in biomineralization were upregulated in the mid-1960s treatment (figure 5b), while a smaller fraction of genes was upregulated in the ambient and/or

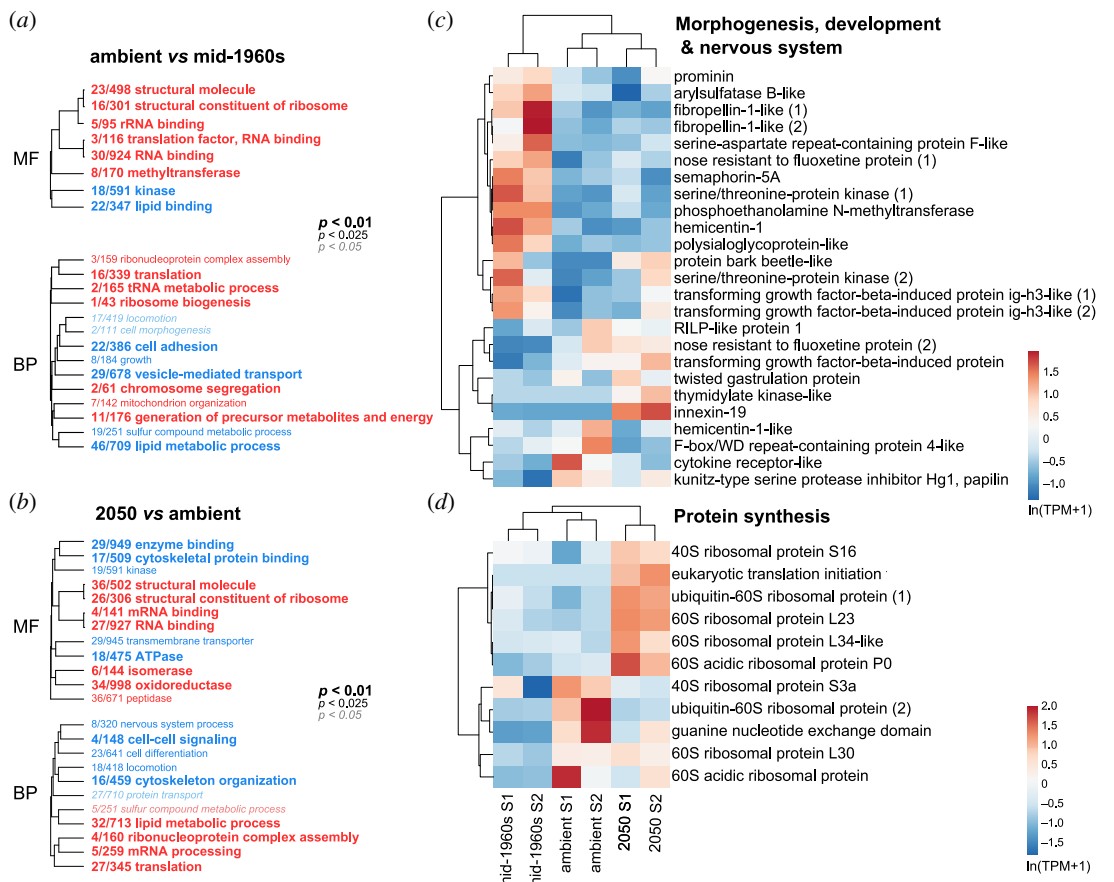

**Figure 4.** Overview of the gene expression response of *Atlanta ariejansseni* juveniles to mid-1960s, ambient and 2050 ocean chemistry conditions. (*a,b*) Hierarchical clustering of gene ontology terms enriched by genes upregulated (red) or downregulated (blue) and summarized by MF and biological process (BP) for each pairwise comparison: (*a*) ambient versus mid-1960s conditions and (*b*) 2050 versus ambient. GO categories associated with protein synthesis were consistently upregulated with decreasing pH: translation (GO:0006412), structural constituents of the ribosome (GO:0003735), RNA binding (GO:0003723), ribosome biogenesis (GO:0042254), and ribonucleoprotein complex assembly (GO:0022618). However, GO categories associated with morphogenesis and organismal development were downregulated with decreasing pH: locomotion (GO:0040011), cell morphogenesis (GO:0000902), cell adhesion (GO:0007155), nervous system process (GO:GO:0050877) and cell differentiation (GO:GO:0030154). The size of the font indicates the significance of the term as indicated by the inset key. The fraction preceding the GO term indicates the number of genes annotated with the term that pass an unadjusted *p*-value threshold of 0.05. Heatmap of the fraction of genes that were responsive to pH changes (adj. *p*-value < 0.05) (*c*) involved in morphogenesis, development and nervous system and (*d*) involved in protein synthesis that were responsive to pH changes (adj. *p*-value < 0.05). Original values of relative abundance of the transcript in units of transcripts per million (TPM) were ln(*x* + 1)-transformed; pareto scaling was applied to rows. Both rows and columns are clustered using correlation distance and mean linkage.

2050 treatments. Biomineralization genes included those coding for candidate extracellular shell matrix proteins such as mucins, perlucin-like and two chitin-binding proteins [76–78], but also genes potentially involved in the transport of proteins and ions to the biomineralization site, including a sodium dependent transporter and a calcium activated-channel regulator [79]. Genes coding for mucins and chitin-binding proteins were strongly upregulated in the mid-1960s treatment, while those coding for perlucins were upregulated in the 2050 treatment.

# 4. Discussion

## 4.1. A complex response to OA

The results of this first study on the effects of OA on atlantid heteropods have revealed a complex organismal response. Although differences in pH between the treatments were relatively small, and

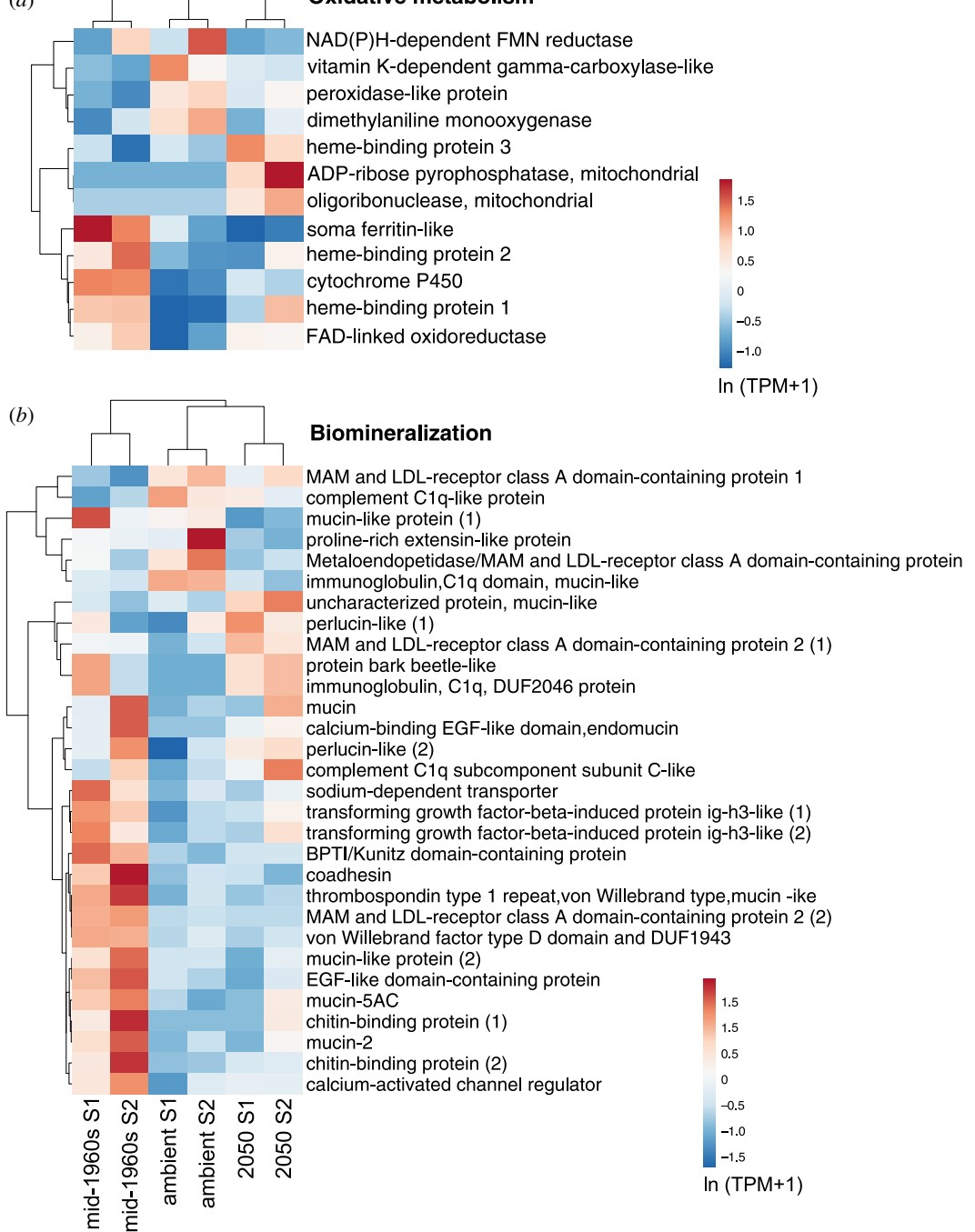

**Figure 5.** Expression patterns of genes of *Atlanta ariejansseni* juveniles in response to mid-1960s, ambient and 2050 ocean chemistry conditions, involved in (*a*) oxidative metabolism and (*b*) biomineralization (adj. *p*-value < 0.05). Original values of relative abundance of the transcript in units of transcripts per million (TPM) were ln($x$ + 1)-transformed; pareto scaling was applied to rows. Both rows and columns are clustered using correlation distance and mean linkage.

aragonite super saturation was maintained across all experiments, the juvenile atlantids responded differently to the different ocean chemistry conditions. These responses demonstrate that, in common with other juvenile molluscs [9,10], the juvenile atlantids are highly sensitive to changing ocean chemistry. Morphological measurements reveal a reduction in shell extension and shell volume from the mid-1960s to the ambient conditions, but an increase in shell extension from ambient to 2050 conditions. At the transcriptomic level, our study reveals some of the most sensitive genes to OA in *Atlanta ariejansseni*. Although different responses were observed for each treatment, the genes found to be differentially expressed are involved in the same BP consistently affected by OA in previous studies on pteropods and other marine calcifiers [75].

Combining these findings indicates two potential responses of atlantids to OA. A first interpretation is that increasing OA negatively affects atlantid calcification. A reduction in shell extension and shell volume from the mid-1960s to the present conditions suggests that ambient water chemistry in the South Atlantic Ocean is already limiting atlantid calcification. A reduction in calcification from the past to the present has also been found in shelled pteropods from time-series data, although the type of calcification response was different to the atlantids. Variation in shell extension cannot be measured from time-series data. Instead, pteropods showed variation in shell thickness, density and porosity [80,81]. Two perturbation experiments to consider past carbonate chemistry (higher pH than ambient) on shelled pteropods found no differences in shell growth between past and ambient conditions [70,82]. However, recently, the pteropod *Limacina retroversa* was shown to produce larger shell volumes under mid-1880s carbonate chemistry conditions when compared to ambient conditions in the sub-Antarctic [24]. A reduction in calcification from the past to the present has been identified in other mollusc groups, and other calcifying planktonic taxa. For example, bivalves grown under pre-industrial conditions were found to grow thicker, more robust shells, as well as growing faster and reaching metamorphosis sooner than those grown under present conditions [83], and planktonic foraminifera from the present-day Southern Ocean were found to weigh 30–35% less than those in the underlying Holocene-aged sediments [84].

An increase in atlantid shell extension from ambient to 2050 conditions may indicate a stress response to grow to a larger size as quickly as possible under increasing OA. It has been shown that some calcifying organisms are able to increase the rates of BP, such as metabolism and calcification, in response to low pH in order to compensate for the negative effects that increased acidity can cause, such as impaired calcification [85,86]. However, this often comes at a cost to overall fitness [85] and such increased rates cannot be sustained in the long term. Our gene expression results indicate that genes involved in protein synthesis are consistently upregulated in one direction with decreasing pH (from mid-1960s to ambient to future 2050).

A second interpretation of our results is that juvenile *Atlanta ariejansseni* are well adapted to ambient carbonate chemistry and any deviation away from this (either decreased pH or increased pH) results in a stress response of increased shell growth rate. Gene expression results show that several genes are not upregulated or downregulated in a linear way with changing pH, suggesting that moving away from ambient carbonate chemistry impacts BP, resulting in a distinct molecular and phenotypic response. For example, biomineralization genes coding for mucins and chitin-binding proteins were strongly upregulated in the mid-1960s treatment, while those coding for perlucins were upregulated in the 2050 treatment. Similar genes were also found to be differentially expressed in pteropod species under low pH treatments [22,23,72,87]. When combined with the variations in shell measurement, these transcriptional differences in biomineralization genes indicate that distinct genes underlie shell extension and shell thickness.

Both interpretations indicate that atlantids are likely to be negatively affected by imminent OA. However, results of our OA and ambient growth experiments suggest that food availability also has an important impact on atlantid calcification, and it may be that increased food supply can mitigate some of the negative effects of OA on juvenile atlantids. For all specimens measured, the shell produced during the OA experiments was thicker than the shell grown *in situ* prior to the experiment. This enhanced calcification throughout the experiments is likely related to the availability of plentiful food in all treatments. OA experiments on shelled pteropods indicate that a lack of food has negative effects on specimen condition [46]. Nutrition is also likely important for atlantid shell production because they are thought to calcify close to the deep chlorophyll maximum [12], a region of higher food availability. Therefore, the juvenile atlantids in OA incubations were all kept under *ad libitum* food conditions. Specimens were observed to feed well during the experiments, with the chlorophyll-rich algae in their stomachs clearly visible through their transparent shells (electronic supplementary material, figure S2). All of the 3-day OA experimental treatments received the same amount of food, so the effects of food are independent from the effects of the three different ocean chemistry conditions in our experiment. In the separate ambient growth experiment that received an eight times higher food concentration, a further 1.62 times increase in the shell extension was found when compared to the ambient OA experiment.

These results suggest that increased food availability leads to increased calcification, and that a plentiful food supply could offset OA induced reduction in calcification. Similar trends have been found in other calcifying organisms, including pteropods [28], corals [88,89] and the early benthic stages of the bivalve mollusc *Mytilus edulis* [90]. The pteropod *Heliconoides inflatus* was found to produce shells that were 40% thicker and 20% larger in diameter during periods of naturally high

nutrient concentrations in the Cariaco basin (compared to specimens sampled during oligotrophic conditions) [28]. A review of OA studies on calcifying marine organisms found that an intermediate or high food supply under laboratory and field conditions increased the resistance to low pH for growth and calcification [91]. At the adult stage, however, atlantids feed primarily on shelled pteropods [3], and the assumed decline in the abundance of OA sensitive pteropods [92] will have a negative effect on food availability for adult atlantids.

The fairly short time that it takes an atlantid to reach maturity may mean that multiple generations are produced each year, and this could help atlantids adapt more quickly to a rapidly changing ocean [93]. Recent research indicates that calcifying organisms can gain tolerance to, and increase physical resistance to OA over several generations [94,95]. Coralline algae were found to become tolerant to lowered pH in only six generations [95], and benthic gastropods in naturally $CO_2$ enriched waters built more durable shells over multiple generations [94]. The timing of the atlantid life cycle is, until now, completely unknown and our results give a first estimate of the minimum longevity of *Atlanta ariejansseni*. If atlantids reach reproductive maturity in approximately 116 days, this could allow for more than one generation per year. This is comparable to the shelled pteropods, which are thought to live for approximately 1–2 years and may produce two generations of offspring per year in the Southern Ocean [96,97], and in more temperate waters [98]. During specimen collection in the South Atlantic Ocean for the present study, small juvenile and large adult specimens were present in the same location at the same time, supporting this inference. Juvenile specimens of *A. ariejansseni* have been caught in sediment traps moored in the Southern Ocean offshore of Tasmania (47° S, 142° E) at the beginning (September) and end (February) of the summer growing season [92]. This suggests that, similar to the pteropod *Limacina helicina antarctica* [96,97], *A. ariejansseni* could have an overwintering juvenile population.

In summary, the findings of this study indicate that calcification in atlantids is likely to be affected by future OA. However, some of the effects of OA on atlantid calcification could be mitigated by plentiful food. Our results demonstrate that the effects of OA on atlantid calcification are not straight forward, and likely depend on whether these organisms are able to survive and maintain calcification under stressful conditions in the long term. Evidence suggests that both shelled pteropods and atlantids survived the Cretaceous-Paleogene extinction event (KPg or KT) and Paleocene-Eocene thermal maximum (PETM), both periods of extreme perturbation in the ocean's carbon cycle [38,39]. These findings give some hope that aragonite shelled holoplanktonic gastropods will be able to adapt to our changing oceans, even though the rate of change is unprecedented relative to the geological record. *Atlanta ariejansseni* resides in cool convergence waters where rapid changes in water temperature and stratification are expected to be additional stressors [99]. Future studies should seek to understand the synergistic effects of OA and warming, to understand variability in the environment in which atlantids live (and environmental tolerances that they may already have), and to thoroughly investigate how nutrition affects calcification and growth.

Data accessibility. All data supporting the findings of this study is provided in the online electronic supplementary material [100]. Raw reads used in this study were deposited at NCBI BioProject PRJNA590142. The Transcriptome Shotgun Assembly has been deposited at DDBJ/EMBL/GenBank under the accession GIOD00000000. The version described in this paper is the first version, GIOD01000000.

Authors' contributions. D.W.-P., L.M., K.T.C.A.P. and P.R.S. designed the study, D.W.-P., L.M., K.T.C.A.P. and E.G. performed the research, D.W.-P., L.K.D., K.B., P.R.S. and E.D. carried out sample preparation and analysis. D.W.-P., L.M. and P.R.S. carried out data analysis. All authors contributed to manuscript preparation.

Competing interests. The authors declare no competing interests.

Funding. This project has received funding from the European Union's Horizon 2020 research and innovation programme under the Marie Sklodowska-Curie grant agreement no. 746186 [POSEIDoN, DW-P] and grant agreement no. 844345 [EPIC, PRS]. Plankton collection on the AMT27 cruise was funded by a Vidi grant no. (016.161351) from the Netherlands Organisation for Scientific Research (NWO) to KTCAP. The Atlantic Meridional Transect is funded by the UK Natural Environment Research Council through its National Capability Long-term Single Centre Science Programme, Climate Linked Atlantic Sector Science (grant no. NE/R015953/1). This study contributes to the international IMBeR project and is contribution number 335 of the AMT programme. L.K.D. was supported by the Netherlands Earth System Science Centre (NESSC), grant no. 024.002.001 from the Dutch Ministry of Education, Culture and Science.

Acknowledgements. We are very grateful to Rob Langelaan and Dirk van der Marel (Naturalis) for microCT scanning specimens, to Vassilis Kitidis (Plymouth Marine Laboratory) and Matthew Humphreys (NIOZ) for discussion on carbonate chemistry and checking carbonate system calculations, and to the captain, crew and scientists who took part in cruise DY084/085 (AMT27) onboard the *RRS Discovery* (PSO: A. Rees). We are also grateful to three anonymous reviewers for providing valuable comments and suggestions on our manuscript.

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
