## [Peer Review File · Royal Society Open Science]

Review History

RSOS-202265.R0 (Original submission)

Review form: Reviewer 1 (James McClintock)

Is the manuscript scientifically sound in its present form?

Yes

Are the interpretations and conclusions justified by the results?

Yes

Is the language acceptable?

Yes

Do you have any ethical concerns with this paper?

No

Have you any concerns about statistical analyses in this paper?

No

Recommendation?

Accept with minor revision (please list in comments)

Comments to the Author(s)

See attached (Appendix A).

Review form: Reviewer 2

Is the manuscript scientifically sound in its present form?

No

Are the interpretations and conclusions justified by the results?

Yes

Is the language acceptable?

Yes

Do you have any ethical concerns with this paper?

No

Have you any concerns about statistical analyses in this paper?

Yes

Recommendation?

Major revision is needed (please make suggestions in comments)

Comments to the Author(s)

This experiment examines the morphological and genetic responses of atlantids to pH changes. The study identifies some key potential effects of OA on an organism that is rarely studied, yet ecologically important. As such, the study is novel and uses multiple methods to answer some key questions on the biology of the organism in regard to climate change. While the MS is generally interesting and easy to follow, there are some important points that need to be clarified. These include adding more information about the feeding comparison, revisiting the statistical analyses and restructuring the result/discussion. Also, the MS reads more like a comparison between the biology of heteropods and pteropods. I understand that these are somehow similar but perhaps the MS should focus a bit more on the biology and ecology of the heteropods. Please find attached further comments (see Appendix B).

Review form: Reviewer 3

Is the manuscript scientifically sound in its present form?

No

Are the interpretations and conclusions justified by the results?

No

Is the language acceptable?

Yes

Do you have any ethical concerns with this paper?

Yes

Have you any concerns about statistical analyses in this paper?

No

Recommendation?

Major revision is needed (please make suggestions in comments)

Comments to the Author(s)

I think authors have a nice history, which, for example, can be oriented to describe the current relationship between snail physiology and climate change drivers.

L75-76. Is this an assumption or assertion? If it is the latter, please provide a reference.

L82-87. The function and likely evolution of the shell in pteropods and atlantids are not similar and hence, their physiological responses including gene expression to OA might diverge as well. Authors could mention this analogy.

L87-88. Why should Atlantids be vulnerable to OA?

L90-92 Like most calcifiers...

L96. This is not totally correct. Shell-production during early developmental stages is restricted under elevated CO₂ levels, but shell dissolution does not occur in particular in adult individuals.

L98-101. This ecological strategy of diel migrations, can increase the tolerance of atlantids to high CO₂ instead of undermine it, such has been demonstrated in others zooplankton taxa (Lewis, et al., 2013; Vennello et al., 2018). Authors might also consider this possibility.

L104-109. This paragraph is in line with the above comments.

L111-113. The sentence suggests the reader a discussion of the potential differences in physiological and genetically-mediated processes of pteropods and atlantids under field-laboratory conditions simulating OA and other climate change-related drivers. However, the rest of the paragraph is vaguely connected with the problematic. I suggest a major edition of this section.

L128-142. In connection with the preceding paragraph, I expected to see a discussion or mention of the physical-chemical conditions shaping the habitat of atlantids. High latitude ecosystems are a conveyor for OA, and organisms are already experiencing changes in their habitats. This, the environmental history, is crucial to understand and contextualize inter and intra-specific differences on the adaptive response to climate change

L129 Replace heteropods by atlantids. Transcriptome needs to be presented in the context of its contribution to solve the problem.

L133-134. It would be illustrative to know the physical-chemical conditions shaping the environment of atlantids.

L128-142. I can't recognize a clear and testable hypothesis, neither the associated methods addressed in the current study. For example, for those questions stated between L133-137, how were these questions assessed? For clarity, this information needs to be provided to the readers.

L146-152. In the Introduction section, authors focused shell functionality on adult stages. If the study was conducted with early development stages, information associated with the shell generation, linked or not to specific environmental regulation, should be mentioned in the Introduction section as well. Such as the authors mention in L152-154, these early development stages can show higher sensitivity to deviations in habitat suitability.

L156-158. Whether juveniles are herbivorous, which function does the shell develop on these stages? Is it already present or is under development? Is there any relationship between shell production and environmental drivers? In other similar taxa? This information needs to be provided earlier to the readers.

- L160-167. Given the information provided between L117-125, I was expecting the study was conducted at a high latitude ecosystem placed in the Atlantic basin.
- L175-178. To which depth this water belongs to (ambient water)?
- L180. Which is the rationale of this treatment?
- L182. Add the depth.
- L183. A relatively high and wide temperature range for a temperate ocean region. To which bathymetric range this temperature variation belongs to?
- L184. Mention if pH was measured with an electrode, spectrophotometer or probe, the precision, reference material, uncertainty (revise the Guide of good practices in OA studies).
- L189. Replace natural by wild.
- L189-194. Provide the water sampling depth for chlorophyll measurements and food supply.
- L190-194. I don't see the point of changing the wild food availability if authors are interested in characterizing responses to natural conditions. Whether current pH/pCO₂ ocean levels constitute an environmental driver affecting phenotypic plasticity and genetic diversity of populations, ultimately selecting for adaptation to ocean acidification, the linkage between site-specific pH variability and physiological (for example) responses needs to be demonstrated without artificial perturbations that might hinder such adaptive process (Lewis et al., 2013; McElhany, 2017).
Lewis, C.N. et al., 2013. *Proc. Natl. Acad. Sci.*, 110 (51), E4960-E4967.
McElhany, P. (2017). *ICES Journal of Marine Science*, 74(4), 926-928.
- L203. Were the food conditions and incubation water properties replaced during this period?
- L211-218. This paragraph belongs to the Results section.
- L251-252. This paragraph belongs to the Results section.
- L254. Shell cleaning and fluorescent imaging to... was carried out..
- L286-287. This paragraph belongs to the Results section.
- L309-331. This paragraph belongs to the Methods section. Furthermore, the goals attained by utilizing this methodology and how its results help to test the hypothesis, need to be better and clearer presented within the Introduction section.
- L333-371. Same as above.
- L375. Rephrase. If the wild food environment was modified, these responses are not under ambient conditions.
- L377-382. This paragraph mixed results and discussion and it needs to be edited.
- L384. "over three to eleven days" unclear sentence.
- L386-388. This paragraph mixed results and discussion and it needs to be edited.
- L393-395. This paragraph belongs to the Discussion section.
- L395-397. I couldn't understand this sentence.
- L384-401. Most of the content of this paragraph has little incidence on the assessed treatments.
- L403-415. This complete section has no sense with regards to the form and content of the research article.
- L419-421. According to this rationale, pCO₂/pH variations select both ways the phenotypic plasticity and genetic diversity of populations. Therefore, calcification will increase as pH increases according to a certain pattern. It also assumes the distribution range both meridional and bathymetric of these populations before the industrial revolution, showed similar (and constant) pCO₂ values than the atmosphere. Thus, the role of the physical CO₂ pump operating and capturing atmospheric CO₂ at the Southern Ocean, since the establishment of the modern ocean circulation millions of years ago, and its evolutionary impact on species present even before, seems to be neglected.

Discussion section needs to be synthesized and focused.

Decision letter (RSOS-202265.R0)

Dear Dr Wall-Palmer

The Editors assigned to your paper RSOS-202265 "The impacts of past, present and future ocean chemistry on predatory planktonic snails" have now received comments from reviewers and would like you to revise the paper in accordance with the reviewer comments and any comments from the Editors. Please note this decision does not guarantee eventual acceptance.

Please submit your revised manuscript and required files (see below) no later than 21 days from today's (ie 22-Apr-2021) date. Note: the ScholarOne system will 'lock' if submission of the revision is attempted 21 or more days after the deadline. If you do not think you will be able to meet this deadline please contact the editorial office immediately.

on behalf of Dr Maximilian Telford (Associate Editor) and Pete Smith (Subject Editor)
openscience@royalsociety.org

Associate Editor Comments to Author (Dr Maximilian Telford):

The manuscript has received three very thorough reviews which will help greatly in bringing it to a publishable state. The three reviewers all appreciated the interest of the study but all found quite a number of points that must be carefully addressed in a thoroughly revised version of the manuscript. I found that the points made were fair (although extensive) and can for the most part

be addressed by a careful rewriting without the need of additional experiments (tho some reanalyses are suggested).

Reviewer comments to Author:

Reviewer: 1

Comments to the Author(s)

See attached file (Wall Palmer et al. 3.22.2021.pdf).

Reviewer: 2

Comments to the Author(s)

This experiment examines the morphological and genetic responses of atlantids to pH changes. The study identifies some key potential effects of OA on an organism that is rarely studied, yet ecologically important. As such, the study is novel and uses multiple methods to answer some key questions on the biology of the organism in regard to climate change. While the MS is generally interesting and easy to follow, there are some important points that need to be clarified. These include adding more information about the feeding comparison, revisiting the statistical analyses and restructuring the result/discussion. Also, the MS reads more like a comparison between the biology of heteropods and pteropods. I understand that these are somehow similar but perhaps the MS should focus a bit more on the biology and ecology of the heteropods. Please find attached further comments ("Review of Wall-Palmer et al.pdf").

Reviewer: 3

Comments to the Author(s)

I think authors have a nice history, which, for example, can be oriented to describe the current relationship between snail physiology and climate change drivers.

L75-76. Is this an assumption or assertion? If it is the latter, please provide a reference.

L82-87. The function and likely evolution of the shell in pteropods and atlantids are not similar and hence, their physiological responses including gene expression to OA might diverge as well. Authors could mention this analogy.

L87-88. Why should Atlantids be vulnerable to OA?

L90-92 Like most calcifiers...

L96. This is not totally correct. Shell-production during early developmental stages is restricted under elevated CO₂ levels, but shell dissolution does not occur in particular in adult individuals.

L98-101. This ecological strategy of diel migrations, can increase the tolerance of atlantids to high CO₂ instead of undermine it, such has been demonstrated in others zooplankton taxa (Lewis, et al., 2013; Vennello et al., 2018). Authors might also consider this possibility.

L104-109. This paragraph is in line with the above comments.

L111-113. The sentence suggests the reader a discussion of the potential differences in physiological and genetically-mediated processes of pteropods and atlantids under field-laboratory conditions simulating OA and other climate change-related drivers. However, the rest of the paragraph is vaguely connected with the problematic. I suggest a major edition of this section.

L128-142. In connection with the preceding paragraph, I expected to see a discussion or mention of the physical-chemical conditions shaping the habitat of atlantids. High latitude ecosystems are a conveyor for OA, and organisms are already experiencing changes in their habitats. This, the environmental history, is crucial to understand and contextualize inter and intra-specific differences on the adaptive response to climate change

L129 Replace heteropods by atlantids. Transcriptome needs to be presented in the context of its contribution to solve the problem.

L133-134. It would be illustrative to know the physical-chemical conditions shaping the environment of atlantids.

L128-142. I can't recognize a clear and testable hypothesis, neither the associated methods addressed in the current study. For example, for those questions stated between L133-137, how were these questions assessed? For clarity, this information needs to be provided to the readers.

L146-152. In the Introduction section, authors focused shell functionality on adult stages. If the study was conducted with early development stages, information associated with the shell generation, linked or not to specific environmental regulation, should be mentioned in the Introduction section as well. Such as the authors mention in L152-154, these early development stages can show higher sensitivity to deviations in habitat suitability.

L156-158. Whether juveniles are herbivorous, which function does the shell develop on these stages? Is it already present or is under development? Is there any relationship between shell production and environmental drivers? In other similar taxa? This information needs to be provided earlier to the readers.

L160-167. Given the information provided between L117-125, I was expecting the study was conducted at a high latitude ecosystem placed in the Atlantic basin.

L175-178. To which depth this water belongs to (ambient water)?

L180. Which is the rationale of this treatment?

L182. Add the depth.

L183. A relatively high and wide temperature range for a temperate ocean region. To which bathymetric range this temperature variation belongs to?

L184. Mention if pH was measured with an electrode, spectrophotometer or probe, the precision, reference material, uncertainty (revise the Guide of good practices in OA studies).

L189. Replace natural by wild.

L189-194. Provide the water sampling depth for chlorophyll measurements and food supply.

L190-194. I don't see the point of changing the wild food availability if authors are interested in characterizing responses to natural conditions. Whether current pH/pCO₂ ocean levels constitute an environmental driver affecting phenotypic plasticity and genetic diversity of populations, ultimately selecting for adaptation to ocean acidification, the linkage between site-specific pH variability and physiological (for example) responses needs to be demonstrated without artificial perturbations that might hinder such adaptive process (Lewis et al., 2013; McElhany, 2017).

Lewis, C.N. et al., 2013. *Proc. Natl. Acad. Sci.*, 110 (51), E4960-E4967.

McElhany, P. (2017). *ICES Journal of Marine Science*, 74(4), 926-928.

L203. Were the food conditions and incubation water properties replaced during this period?

L211-218. This paragraph belongs to the Results section.

L251-252. This paragraph belongs to the Results section.

L254. Shell cleaning and fluorescent imaging to... was carried out..

L286-287. This paragraph belongs to the Results section.

L309-331. This paragraph belongs to the Methods section. Furthermore, the goals attained by utilizing this methodology and how its results help to test the hypothesis, need to be better and clearer presented within the Introduction section.

L333-371. Same as above.

L375. Rephrase. If the wild food environment was modified, these responses are not under ambient conditions.

L377-382. This paragraph mixed results and discussion and it needs to be edited.

L384. "over three to eleven days" unclear sentence.

L386-388. This paragraph mixed results and discussion and it needs to be edited.

L393-395. This paragraph belongs to the Discussion section.

L395-397. I couldn't understand this sentence.

L384-401. Most of the content of this paragraph has little incidence on the assessed treatments.

L403-415. This complete section has no sense with regards to the form and content of the research article.

L419-421. According to this rationale, pCO₂/pH variations select both ways the phenotypic plasticity and genetic diversity of populations. Therefore, calcification will increase as pH increases according to a certain pattern. It also assumes the distribution range both meridional and bathymetric of these populations before the industrial revolution, showed similar (and constant) pCO₂ values than the atmosphere. Thus, the role of the physical CO₂ pump operating and capturing atmospheric CO₂ at the Southern Ocean, since the establishment of the modern ocean circulation millions of years ago, and its evolutionary impact on species present even before, seems to be neglected.

Discussion section needs to be synthesized and focused.

===PREPARING YOUR MANUSCRIPT===

===PREPARING YOUR REVISION IN SCHOLARONE===

Author's Response to Decision Letter for (RSOS-202265.R0)

See Appendix C.

RSOS-202265.R1 (Revision)

Review form: Reviewer 1 (James McClintock)

Is the manuscript scientifically sound in its present form?

Yes

Are the interpretations and conclusions justified by the results?

Yes

Is the language acceptable?

Yes

Do you have any ethical concerns with this paper?

No

Have you any concerns about statistical analyses in this paper?

No

Recommendation?

Accept as is

Comments to the Author(s)

I have carefully reviewed your point by point edits to your revised manuscript in response to the detailed editorial recommendations of your three reviewers. I am pleased with your response to my recommendations, and it seems to me that you have adequately addressed the concerns and recommendations of the other two reviewers in your revised manuscript. I am pleased to recommend publication.

Review form: Reviewer 2

Is the manuscript scientifically sound in its present form?

Yes

Are the interpretations and conclusions justified by the results?

Yes

Is the language acceptable?

Yes

Do you have any ethical concerns with this paper?

No

Have you any concerns about statistical analyses in this paper?

No

Recommendation?

Accept as is

Comments to the Author(s)

The revision has addressed the comments and adjusted the manuscript well. This is a nice MS and should be interesting for the readers.

Decision letter (RSOS-202265.R1)

Dear Dr Wall-Palmer,

It is a pleasure to accept your manuscript entitled "The impacts of past, present and future ocean chemistry on predatory planktonic snails" in its current form for publication in Royal Society Open Science. The comments of the reviewer(s) who reviewed your manuscript are included at the foot of this letter.

on behalf of Prof Pete Smith (Subject Editor)
openscience@royalsociety.org

Associate Editor Comments to Author:
Congratulations on the acceptance of your manuscript!

Reviewer comments to Author:

Reviewer: 2

Comments to the Author(s)

The revision has addressed the comments and adjusted the manuscript well. This is a nice MS and should be interesting for the readers.

Reviewer: 1

Comments to the Author(s)

I have carefully reviewed your point by point edits to your revised manuscript in response to the detailed editorial recommendations of your three reviewers. I am pleased with your response to my recommendations, and it seems to me that you have adequately addressed the concerns and recommendations of the other two reviewers in your revised manuscript. I am pleased to recommend publication.

Appendix A

This is a timely combination of a review and an original experimental study examining the interplay between ocean acidification and planktonic snails. The implications of the impacts of growing ocean acidification on these key plankton are considerable. Shelled planktonic snails are so abundant that their sinking shells have a seafloor substrate named for them (pteropod ooze) and they contribute to the global carbon cycle. As such, understanding their fate in terms of shell dissolution and the rising subsaturation levels of aragonite, with which they build their delicate shells, is critical.

The following comments may be of assistance in preparing a minor revision:

Line 65. Give the shell dimension of the 14 mm. Readers won't know.

Line 70. I had no idea that some use their shell as a swimming appendage. Briefly explain how so.

Line 75. '...in particular rapid contemporary changes in ocean chemistry.'

Line 87. Consider adding the following appropriate citation:

Fabry, V.J., **J.B. McClintock**, J.T. Mathis and J.M. Grebmeir. 2009. Ocean acidification at high latitudes: The Bellwether. *Oceanography* 22, 160-171.

Line 100. Consider adding the following appropriate citation:

Lebrato, M., A.J. Andersson, J.B. Ries, R.B. Aronson, M.D. Lamare, W. Koeve, A. Oshlies, M.D. Iglesias-Rodriguez, S. Thatje, M. Amsler, S.C. Vos, D., O.B. Jones, H.A. Ruhl, A.R. Gates, and **J.B. McClintock**. 2016. Benthic calcifiers coexist with seawater-carbonate undersaturation at a global scale. *Global Biogeochemical Cycles* 30, 1038- 1053.

Line 104. Might also be a place to cite Fabry et al. 2009.

Line 121. '...where this species can reach...'

Line 157. '...with an algal food source...'

Line 157. So if I follow this correctly food was only provided at the beginning of the experiment but not during the experimental period. This is unclear. If it is true, any consequences? Does this species feed continuously in the plankton in nature?

Line 178. Good place to give a brief explanation and justification of choice of pH levels. The level chosen at first glance does not appear very aggressive. Yet in retrospect, this worked well given the impacts detected.

Line 186 and 187. So it seems that the carboys maintained their pH despite not being bubbled with CO₂ and air?

Line 191. Was the 'dry algae' powdered?

Line 193. What was the basis of the amount of food added each day? Ad libitum?

Line 198. '...each of the three treatments.'

Line 237. From carboys?

Line 254. Shell 'extension' sounds odd. Maybe define extension the first time it is used?

Line 274. Pg 8. When the term 'random' is used you need to actually use statistics to carry this out. Perhaps you mean 'haphazard'?

Line 311. How many samples?

Line 400. '...specimen to grow from ? to full adult size...'

Line 411. "'caught in sampling nets at the...'

Line 414. Add citation after *antarctica*

Line 566. Maybe use *ad libitum* instead of 'repleat'

Lines 590-592. Any of these studies done in the field or all lab based? Might mention.

Line 621. Any field studies you can cite that have found higher resistance to OA (less in situ shell dissolution) in shelled pteropods in areas where plankton (food) is higher than control areas with lower levels of plankton (food)?

Appendix B

Review of 'The impacts of past, present and future ocean chemistry on predatory planktonic snails'

General comments:

This experiment examines the morphological and genetic responses of atlantids to pH changes. The study identifies some key potential effects of OA on an organism that is rarely studied, yet ecologically important. As such, the study is novel and uses multiple methods to answer some key questions on the biology of the organism in regard to climate change. While the MS is generally interesting and easy to follow, there are some important points that need to be clarified. These include adding more information about the feeding comparison, revisiting the statistical analyses and restructuring the result/discussion. Also, the MS reads more like a comparison between the biology of heteropods and pteropods. I understand that these are somehow similar but perhaps the MS should focus a bit more on the biology and ecology of the heteropods. Please see below for further comments.

Abstract

The abstract seems slightly vague, especially regarding the key results found in the study. It needs to bring forward the key findings of the study (including the transcriptomics) rather than just stating that there were differences in response between past, present and future treatments. What are the key findings of this study in terms of the physiology and why are they so important?

Introduction

Methods

Lines 175 to 194: from the way the information is presented, it first sound like only one replicate tank was being used per treatment. Perhaps a clearer way to explain the experimental design is to state what the treatments were, the number of replicates per treatment, and then mention that the water for each treatment was pre-mixed by bubbling.....

Also, it sounds like only two replicate carboys were used per treatment. If so, why?

Lines 291 to 293.: It is not clear why a test for categorical data was used to test correlations.

Line 297: Levene's test is used to test for homoscedasticity. Why not use a Kolmogorov-Smirnov test or Shapiro-Wilk test to test for normality?

Line 302: do the authors mean Kruskal-Wallis?

Line 305: Why was a pair-wise comparison of the Kruskal-Wallis test not used?

I would strongly suggest revisiting the statistical analyses and perhaps analysing the differences between data sets using PERMANOVA. This will be more robust and avoid the need to use different tests to analyse data based on their distribution. The authors can then add a table of the PERMANOVA results, providing all information (degrees of freedom, etc). The table can then be referenced to when describing differences/similarities in the data.

Results/Discussion

This section seems a bit long. In addition, the last part (a complex response to OA), sounds like a repeat of the previously discussed ideas within the section. I understand that it is a summary of the complexity in response and provides some of the key points. It actually reads well and does a good job at pointing out the main discussion points. I would suggest separating the results and discussion. In my point of view, it would provide a clearer understanding of the findings and a clearer discussion of findings. The last part of the discussion can be expanded with the parts before it and possibly adding further details on the physiology (perhaps comparing trade-offs) using the transcriptomics data.

Also, the section about feeding seems to come from nowhere, since it is not mentioned in the introduction or methods. It sounds like a quick observation that was thrown in the mix to add to the MS.

The referencing of statistical results used is not very uniform. Again, perhaps using a PERMANOVA and providing a table of the results will be more robust and uniform.

The variability observed in some of the shell data is quite interesting. What does this mean in terms of the genetic diversity or phenotypes?

Finally, while it is interesting to compare the current findings to those from pteropods, the MS reads a bit like that is the primary goal rather than focussing on the biology and ecology of heteropods and discussing the results generally with other calcifying organisms, including those that only have larvae in the planktonic stage.

Lines 393 to 395: What is meant by 'the atlantid increases with age'? Perhaps this sentence needs restructuring.

Line 399: It is unclear what the number in brackets represent?

Lines 419 to 431: This part of the paragraph reads a lot like a repeat of the methods. Perhaps it would be better to remove it for the results/discussion section and clarify it better in the methods.

Lines 432 to 436: This section may be merged with the following paragraph.

Line 471: remove mean and sd in brackets.

Lines 474 to 478: This sentence is a bit vague and seem to come from the abstract of the cited paper. What do the authors mean by 'in response to increased acidity'? Please specify how.

Line 478: This sentence may be excluded.

Line 490 to 493: This sentence seems to be a rough comparison and does not add information on the study.

Line 521 to 523: 'Were strongly upregulated in the future' sounds odd. Perhaps the authors mean 'in the future treatment'. Also, in some parts of the text the authors refer to '2050 treatments' and in other parts 'future 2050 treatments'. Perhaps this can be standardised throughout the text by referring to the treatments as 'mid-1960 treatments', 'ambient treatments' and '2050 treatments'.

Appendix C

Response to reviewer's comments for the manuscript

'The impacts of past, present and future ocean chemistry on predatory planktonic snails'

We are extremely grateful to the three anonymous reviewers for their thorough reviews of our manuscript. With your insightful comments and suggestions our manuscript is greatly improved. Please see our point-by-point responses to all comments below. **Please note that line numbers included in the responses refer to the clean revised manuscript.**

Reviewer 1

This is a timely combination of a review and an original experimental study examining the interplay between ocean acidification and planktonic snails. The implications of the impacts of growing ocean acidification on these key plankton are considerable. Shelled planktonic snails are so abundant that their sinking shells have a seafloor substrate named for them (pteropod ooze) and they contribute to the global carbon cycle. As such, understanding their fate in terms of shell dissolution and the rising subsaturation levels of aragonite, with which they build their delicate shells, is critical.

The following comments may be of assistance in preparing a minor revision:

Line 65. Give the shell dimension of the 14 mm. Readers won't know.

Line 68: We have changed it to <14 mm diameter.

Line 70. I had no idea that some use their shell as a swimming appendage. Briefly explain how so.

Line 72: We have added more details here so the reader can imagine their swimming technique.

'...use their shell as a swimming appendage, which is paired with their single swimming fin, both being flapped in a coordinated wing-like manner to produce rapid directional propulsion'

Line 75. '...in particular rapid contemporary changes in ocean chemistry.'

Line 80: Text changed as suggested.

Line 87. Consider adding the following appropriate citation:

Fabry, V.J., J.B. McClintock, J.T. Mathis and J.M. Grebmeir. 2009. Ocean acidification at high latitudes: The Bellwether. *Oceanography* 22, 160-171.

Line 92: We have added this citation as suggested.

Line 100. Consider adding the following appropriate citation:

Lebrato, M., A.J. Andersson, J.B. Ries, R.B. Aronson, M.D. Lamare, W. Koeve, A. Oshlies, M.D. Iglesias-Rodriguez, S. Thatje, M. Amsler, S.C. Vos, D.O.B. Jones, H.A. Ruhl, A.R. Gates, and J.B. McClintock. 2016. Benthic calcifiers coexist with seawater carbonate undersaturation at a global scale. *Global Biogeochemical Cycles* 30, 1038- 1053.

Thank you for the suggestion. This paper was an interesting read. We have decided not to include this citation though because it is focussed on Mg-calcite organisms in benthic habitats and we feel this is too distant from our pelagic focus.

Line 104. Might also be a place to cite Fabry et al. 2009.

Line 113: We agree, Fabry et al. 2009 is cited here.

Line 121. ‘...where this species can reach...’

Line 133: Text changed as suggested.

Line 157. ‘...with an algal food source...’

Line 174: Text changed as suggested.

Line 157. So if I follow this correctly food was only provided at the beginning of the experiment but not during the experimental period. This is unclear. If it is true, any consequences? Does this species feed continuously in the plankton in nature?

Line 175: That is correct, the food was added only at the beginning of the experiments in order to reduce disturbance to the animals. We added a replete concentration of food to allow for possible continuous feeding. We have added to the text:

‘Juvenile atlantids likely have periods of active feeding and resting [41]. A replete food supply was added to be sufficient for several days under the possibility that the animals feed near continuously under experimental conditions.’

Line 178. Good place to give a brief explanation and justification of choice of pH levels. The level chosen at first glance does not appear very aggressive. Yet in retrospect, this worked well given the impacts detected.

Line 195: We have moved this explanation from the results and discussion (also on recommendation of Rev. 2). Indeed we intentionally chose realistic recent-past and near-future pH scenarios.

‘To evaluate the effects of ocean carbonate chemistry on atlantid calcification, specimens of *A. ariejansseni* were incubated for three days across realistic recent-past, ambient and near-future pH scenarios. We applied a past scenario of 0.05 pH units higher than ambient (ambient pH 8.14 ± 0.02 , past pH 8.19 ± 0.02 , Table 1) that is approximately equivalent to the mid-1960s (assuming a decrease in pH of 0.001 units per year in this region) [18], and a future OA scenario of 0.11 pH units lower than ambient (pH 8.03 ± 0.00 , Table 1), which is approximately equivalent to expectations for the year 2050 in the South Atlantic Ocean (under IPCC Representative Concentration Pathway RCP8.5) [13,47]. Aragonite saturation was maintained in all scenarios ($\Omega > 1.82$).’

Line 186 and 187. So it seems that the carboys maintained their pH despite not being bubbled with CO₂ and air?

The carboys did maintain their pH very well because they were sealed air-tight at the start of the experiments with no head space, and not opened until the end of the experiments. In addition, the volume of seawater was large in relation to the small number and size of the juvenile atlantids. We have made this section more clear with some re-structuring and we have added other key pieces of information to explain how the carboys maintained their water chemistry. This includes:

Line 240: ‘Carboys were sealed air-tight with no head space and immediately incubated...’

Line 229: ‘Algae was only added at the beginning of the experiment to prevent disturbance of the specimens and to avoid exposing the experimental seawater to the atmosphere.’

Line 191. Was the ‘dry algae’ powdered?

Line 253: The algae was freeze dried. We have added this detail.

Line 193. What was the basis of the amount of food added each day? Ad libitum?

Line 229: This is a very useful point. We have added:

‘Algae was only added at the beginning of the experiment to prevent disturbance of the specimens and to avoid exposing the experimental seawater to the atmosphere. A high concentration (~500-1000 times more than the *in situ* concentration) of algae was therefore added to ensure that *ad libitum* feeding was possible for the duration of the experiment.’

Line 198. ‘...each of the three treatments.’

Line 237: Text changed as suggested.

Line 237. From carboys?

Line 268: We have changed the heading to make it clear that all measurements were taken from the barrels and carboys.

‘Water chemistry (from barrels and carboys).’

Line 254. Shell ‘extension’ sounds odd. Maybe define extension the first time it is used?

Line 285: We have now defined this, as suggested.

‘One method for quantifying calcification is to measure the length of the piece of shell that grew during the experiment e.g. [57], herein described as the shell extension.’

Line 274. Pg 8. When the term ‘random’ is used you need to actually use statistics to carry this out. Perhaps you mean ‘haphazard?’

Line 236: We were not aware of this. We have removed the word ‘random’ and where a replacement was necessary, we have used ‘arbitrary’.

Line 311. How many samples?

Line 334: We have made the number of samples more clear.

‘...RNA was extracted from two samples of 8–10 individuals pooled from each treatment using the RNeasy Plus Micro Kit (QIAGEN). This sampling provided two replicates per treatment (six samples in total)...’

Line 400. ‘...specimen to grow from ? to full adult size...’

Line 439: We have added ‘from hatching’ to this sentence.

‘...it would take around 116 days for an *A. ariejansseni* specimen to grow from hatching to full adult size...’

Line 411. ‘’caught in sampling nets at the...’

Line 664: The specimens were caught in sediment traps. We have restructured this sentence to make it more clear.

‘Juvenile specimens of *A. ariejansseni* have been caught in sediment traps moored in the Southern Ocean offshore of Tasmania (47°S, 142°E) at the beginning (September) and end (February) of the summer growing season’

Line 414. Add citation after antarctica

Line 668: We have added two citations as suggested:

Bednaršek et al. 2012. Population dynamics and biogeochemical significance of *Limacina helicina* antarctica in the Scotia Sea (Southern Ocean).

Hunt et al. 2008. Pteropods in Southern Ocean ecosystems.

Line 566. Maybe use ad libitum instead of ‘repleat’

Lines 233, 627: As suggested, we have now used ‘ad libitum’ to describe the food/feeding.

Lines 590-592. Any of these studies done in the field or all lab based? Might mention.

Line 643: Most of the reviewed studies were carried out in a lab, but one included field observations. We have added this detail:

‘A review of OA studies on calcifying marine organisms found that an intermediate or high food supply under laboratory and field conditions increased the resistance to low pH for growth and calcification [91].’

Line 621. Any field studies you can cite that have found higher resistance to OA (less in situ shell dissolution) in shelled pteropods in areas where plankton (food) is higher than control areas with lower levels of plankton (food)?

Unfortunately, there are no published studies on shelled pteropods that we know of that investigate food availability and effects of OA. The study of Oakes and Sessa (2020) does find that shell thickness (as a measure of calcification) increased during periods of higher food supply (upwelling with increased carbon flux and diatom blooms). However, no link to OA was made, and no comparisons between areas with varying carbonate chemistry.

Please note that line numbers included in the responses refer to the clean revised manuscript.

Reviewer 2

General comments:

This experiment examines the morphological and genetic responses of atlantids to pH changes. The study identifies some key potential effects of OA on an organism that is rarely studied, yet ecologically important. As such, the study is novel and uses multiple methods to answer some key questions on the biology of the organism in regard to climate change. While the MS is generally interesting and easy to follow, there are some important points that need to be clarified. These include adding more information about the feeding comparison, revisiting the statistical analyses and restructuring the result/discussion. Also, the MS reads more like a comparison between the biology of heteropods and pteropods. I understand that these are somehow similar but perhaps the MS should focus a bit more on the biology and ecology of the heteropods. Please see below for further comments.

Abstract

The abstract seems slightly vague, especially regarding the key results found in the study. It needs to bring forward the key findings of the study (including the transcriptomics) rather than just stating that there were differences in response between past, present and future treatments. What are the key findings of this study in terms of the physiology and why are they so important?

Line 34: We have revised the abstract, removing the vague statements and including more details of the key findings.

‘The atlantid heteropods represent the only predatory, aragonite shelled zooplankton. Atlantid shell production is likely to be sensitive to ocean acidification (OA), and yet we know little about their mechanisms of calcification, or their response to changing ocean chemistry. Here we present the first study into calcification and gene expression effects of short-term OA exposure on juvenile atlantids across three pH scenarios: mid-1960’s, ambient, and 2050 conditions. Calcification and gene expression indicate a distinct response to each treatment. Shell extension and shell volume were reduced from mid-1960s to ambient conditions, suggesting that calcification is already limited in today’s South Atlantic. However, shell extension increased from ambient to 2050 conditions. Genes involved in protein synthesis were consistently upregulated, whereas genes involved in organismal development were downregulated with decreasing pH. Biomineralization genes were upregulated in the mid-1960s and 2050 conditions, suggesting that any deviation from ambient carbonate chemistry causes stress, resulting in rapid shell growth. We conclude that atlantid calcification is likely to be negatively affected by future OA. However, we also found that plentiful food increased shell extension and shell thickness, and so synergistic factors are likely to impact the resilience of atlantids in an acidifying ocean.’

Methods

Lines 175 to 194: from the way the information is presented, it first sound like only one replicate tank was being used per treatment. Perhaps a clearer way to explain the experimental design is to state what the treatments were, the number of replicates per treatment, and then mention that the water for each treatment was pre-mixed by bubbling.....

Line 195: We have thoroughly revised this section as suggested, also moving methods details from elsewhere in the manuscript.

‘To evaluate the effects of ocean carbonate chemistry on atlantid calcification, specimens of *A. ariejansseni* were incubated for three days across realistic recent-past, ambient and near-future pH scenarios. We applied a past scenario of 0.05 pH units higher than ambient (ambient pH 8.14 ± 0.02 , past pH 8.19 ± 0.02 , Table 1) that is approximately equivalent to the mid-1960s (assuming a decrease in pH of 0.001 units per year in this region) [18], and a future OA scenario of 0.11 pH units lower than ambient (pH 8.03 ± 0.00 , Table 1), which is approximately equivalent to expectations for the year 2050 in the South Atlantic Ocean (under IPCC Representative Concentration Pathway RCP8.5) [13,47]. Aragonite saturation was maintained in all scenarios ($\Omega > 1.82$).

Three replicate experiments were conducted for each of the three treatments (n=9, plus three controls, total n=12 carboys). To ensure consistency of water chemistry across the replicates of each treatment, water was pre-treated in barrels before being used to fill experiment carboys. Surface seawater (from ~10 m water depth) was filtered at 0.2 μm into four 60 litre

barrels, which underwent the following treatments. In one barrel, lowered pH was achieved by bubbling...’

Also, it sounds like only two replicate carboys were used per treatment. If so, why?

Line 222: There are three replicate carboys per OA treatment. Unfortunately water chemistry analyses failed for one of the control carboys, so we originally only reported two controls in the manuscript. To make this more clear we have revised the number of control carboys to three, but have added a note about the failed analyses.

‘Three further carboys were filled with ambient seawater to act as controls (no specimens added; water chemistry analyses were successful for two control carboys).’

Lines 291 to 293.: It is not clear why a test for categorical data was used to test correlations.

Line 320: We have checked these calculations and realise that we used the Pearson’s r correlation coefficient (which is the correct one) and not the Pearson’s Chi-squared (which is written in the manuscript). We have corrected the text.

Line 297: Levene’s test is used to test for homoscedasticity. Why not use a Kolmogorov-Smirnov test or Shapiro-Wilk test to test for normality?

Line 302: do the authors mean Kruskal-Wallis?

Line 305: Why was a pair-wise comparison of the Kruskal-Wallis test not used?

I would strongly suggest revisiting the statistical analyses and perhaps analysing the differences between data sets using PERMANOVA. This will be more robust and avoid the need to use different tests to analyse data based on their distribution. The authors can then add a table of the PERMANOVA results, providing all information (degrees of freedom, etc). The table can then be referenced to when describing differences/similarities in the data.

Line 323, Results section and Tables 3 and 4: In response to the four comments above: Following the advice of Reviewer 2, we have revised the statistical analyses, using a one-way PERMANOVA instead of the combination of one-way ANOVA and Kruskal-Wallis tests (and corresponding post-hoc tests). The PERMANOVA results agree with the previously used tests, so the results have not changed. We agree that the PERMANOVA is a much more elegant test, avoiding the need for multiple different tests and avoiding tests for normality. Thank you for this very useful suggestion. Please see the results section for the changes. We have also included two new tables (Tables 3 and 4) of the PERMANOVA results.

Results/Discussion

This section seems a bit long. In addition, the last part (a complex response to OA), sounds like a repeat of the previously discussed ideas within the section. I understand that it is a summary of the complexity in response and provides some of the key points. It actually reads well and does a good job at pointing out the main discussion points. I would suggest separating the results and discussion. In my point of view, it would provide a clearer understanding of the findings and a clearer discussion of findings. The last part of the discussion can be expanded with the parts before it and possibly adding further details on the physiology (perhaps comparing trade-offs) using the transcriptomics data.

Line 397 Results, Line 552 Discussion: We have now restructured the manuscript to separate the results and discussion sections. Thank you for your suggestion to use the last section of the manuscript as the discussion. We have also adopted this change, expanding that section with some of the discussion text moved from the originally combined results and discussion.

Also, the section about feeding seems to come from nowhere, since it is not mentioned in the introduction or methods. It sounds like a quick observation that was thrown in the mix to add to the MS.

Line 92, Line 480: Thank you for this remark. We have added details to the introduction (see below), and we have made a separate results section (The effects of differing food concentrations).

‘The availability of food has also been found to influence pteropod calcification, with food deprivation acting as a synergistic stressor in OA experiments [27], and higher *in situ* food concentrations coinciding with thicker and larger pteropod shells in natural populations [28].’

The referencing of statistical results used is not very uniform. Again, perhaps using a PERMANOVA and providing a table of the results will be more robust and uniform.

Line 323, Results section and Tables 3 and 4: Please see comment above. We have now used PERMANOVA and we have added two tables of the results.

The variability observed in some of the shell data is quite interesting. What does this mean in terms of the genetic diversity or phenotypes?

We agree that understanding the link between genetic diversity and phenotype would be interesting. However, in this study we cannot link the genotype to phenotype directly, as different individuals are used for the gene expression and shell measurements (the shell dissolves in RNA later which is used to preserve the RNA). Also, for the differential gene expression analyses we used a pooled set of individuals to 1) have enough material to sequence RNA, and 2) remove some of the variability in transcriptomic responses of different individuals.

Finally, while it is interesting to compare the current findings to those from pteropods, the MS reads a bit like that is the primary goal rather than focussing on the biology and ecology of heteropods and discussing the results generally with other calcifying organisms, including those that only have larvae in the planktonic stage.

We have included a lot of comparisons to shelled pteropods because they are the most comparable organisms (particularly for the transcriptomic results), and a lot of OA research has been carried out on them. However, we agree that the results should be discussed in a wider context. We have reduced and updated comparisons to pteropods within the discussion, and we have added comparisons to more varied taxa. For example:

Line 583: ‘A reduction in calcification from the past to the present has been identified in other mollusc groups, and other calcifying planktonic taxa. For example, bivalves grown under pre-industrial conditions were found to grow thicker, more robust shells, as well as growing faster and reaching metamorphosis sooner than those grown under present conditions [83], and planktonic foraminifera from the present day Southern Ocean were found to weigh 30–35% less than those in the underlying Holocene-aged sediments [84].’

Lines 393 to 395: What is meant by ‘the atlantid increases with age’? Perhaps this sentence needs restructuring.

Line 431: We have clarified this by adding to the text:

‘This pattern may be due to the relatively broader surface of the shell as the shell shape inflates with increasing age and number of shell whorls, such that the amount of shell produced may be approximately the same.’

Line 399: It is unclear what the number in brackets represent?

Line 437: We have removed the square brackets from this equation. It is now:
($\ln -26.89 \times \text{days} + 95.076$)

Lines 419 to 431: This part of the paragraph reads a lot like a repeat of the methods. Perhaps it would be better to remove it for the results/discussion section and clarify it better in the methods.

Line 195: Thank you for the suggestion. This part of the paragraph has been moved to the methods.

Lines 432 to 436: This section may be merged with the following paragraph.

Line 557: We have made this suggested change.

Line 471: remove mean and sd in brackets.

Line 464: We have removed this and directed the reader to Table 1, where this information is shown.

Lines 474 to 478: This sentence is a bit vague and seem to come from the abstract of the cited paper. What do the authors mean by 'in response to increased acidity'? Please specify how.

Line 593: We have added to this sentence:

'It has been shown that some calcifying organisms are able to increase the rates of biological processes, such as metabolism and calcification, in response to low pH in order to compensate for the negative effects that increased acidity can cause, such as impaired calcification [85,86].'

Line 478: This sentence may be excluded.

We agree, this is discussed in the gene expression section and does not add anything here. We have removed this sentence.

Line 490 to 493: This sentence seems to be a rough comparison and does not add information on the study.

Line 501: We agree that this is just a rough comparison. However, we prefer to keep this sentence because it highlights how many differentially expressed genes are found in relation to the transcriptome size. Moreover, it shows that we have a similar fraction of differentially expressed genes when compared to other zooplankton studies.

Line 521 to 523: 'Were strongly upregulated in the future' sounds odd. Perhaps the authors mean 'in the future treatment'. Also, in some parts of the text the authors refer to '2050 treatments' and in other parts 'future 2050 treatments'. Perhaps this can be standardised throughout the text by referring to the treatments as 'mid-1960 treatments', 'ambient treatments' and '2050 treatments'.

Line 533: We have changed this as suggested. Yes it should have been 'in the future treatment'. As suggested, we have standardised the text and figures regarding the 2050 treatments.

Please note that line numbers included in the responses refer to the clean revised manuscript.

Reviewer 3

I think authors have a nice history, which, for example, can be oriented to describe the current relationship between snail physiology and climate change drivers.

L75-76. Is this an assumption or assertion? If it is the latter, please provide a reference.

Line 78: This is an assumption that is justified later in the introduction. We feel that ‘...despite likely being affected...’ shows that this is an assumption, but we have added a citation to previous work on atlantids that also discusses this. In response to another comment, we have also mentioned larval molluscs and added relevant citations in this section.

‘However, almost nothing is known about the shell structure of atlantids and their mechanisms of calcification [7,8], despite likely being affected by imminent ocean changes, in particular rapid contemporary changes in ocean chemistry, and especially in the early life stages [9–12].’

L82-87. The function and likely evolution of the shell in pteropods and atlantids are not similar and hence, their physiological responses including gene expression to OA might diverge as well. Authors could mention this analogy.

Line 121: We have acknowledged this by adding to the introduction:

‘Although superficially similar, shelled pteropods and atlantid heteropods are phylogenetically and ecologically very distinct, having evolutionarily independent origins and occupying different trophic levels. These differences could mean that they have very different physiological responses to changing ocean carbonate chemistry.’

L87-88. Why should Atlantids be vulnerable to OA?

We elucidate this in the paragraph immediately following this sentence. The explanation starts on Line 100: ‘First, the shell on which atlantids rely is constructed of aragonite...’

L90-92 Like most calcifiers...

Even though most marine calcifiers are likely to be affected by OA, they are extremely diverse (even just within the Mollusca), inhabiting a wide range of habitats (e.g. benthos, plankton, intertidal), producing different forms of calcium carbonate (e.g. calcite, aragonite) and having different tolerances to changing ocean chemistry. This means that not all calcifiers are equally susceptible to OA, and therefore, we have not changed the text as suggested. Even within the plankton, the planktonic foraminifera, for example, are more resistant to OA because their shells/tests are calcite (and they have complex mechanisms of calcification), compared to the aragonite shelled pteropods and atlantids.

L96. This is not totally correct. Shell-production during early developmental stages is restricted under elevated CO₂ levels, but shell dissolution does not occur in particular in adult individuals.

Line 103: While there are no published studies that investigate the effects of OA on atlantids (this manuscript is the first such study), several studies have shown that the shells of the morphologically similar adult and juvenile shelled pteropods undergo dissolution in waters

undersaturated with respect to aragonite. It is thought that the periostracum can protect the pteropod shell from dissolution, but only as long as it remains intact. In areas where the periostracum is damaged, dissolution can occur. The citation on Line 103 is for a study on juvenile pteropods. We have now added a citation for adult pteropods to demonstrate this.

Bednaršek N *et al.* 2012 Extensive dissolution of live pteropods in the Southern Ocean. *Nature Geoscience* 5, 881–885. (doi:10.1038/ngeo1635)

L98-101. This ecological strategy of diel migrations, can increase the tolerance of atlantids to high CO₂ instead of undermine it, such has been demonstrated in others zooplankton taxa (Lewis, et al., 2013; Vennello et al., 2018). Authors might also consider this possibility.

L104-109. This paragraph is in line with the above comments.

Line 109: This is a useful addition to the manuscript, and we agree that diel vertical migrations can increase the tolerance. Indeed, migrating shelled pteropods have been shown to be more tolerant to acidification. We have added:

‘...altered ocean chemistry across the vertical extent of their distributions. However, vertical migration could also help to increase the tolerance of atlantids to OA, as has been shown for some migrating pteropods and other zooplankton groups [35–37].’

L111-113. The sentence suggests the reader a discussion of the potential differences in physiological and genetically-mediated processes of pteropods and atlantids under field-laboratory conditions simulating OA and other climate change-related drivers. However, the rest of the paragraph is vaguely connected with the problematic. I suggest a major edition of this section.

This introductory sentence only states that there are phylogenetic and ecological differences between shelled pteropods and atlantids. It is important to point out the different feeding strategies between pteropods and atlantids in the following text, because many more studies have focussed on the shelled pteropods. As mentioned in the introduction, there are no other OA studies on atlantids, in fact there is only one other study (from 1970) that describes any laboratory work on them at all. We feel it is important to keep this sentence where it is.

L128-142. In connection with the preceding paragraph, I expected to see a discussion or mention of the physical-chemical conditions shaping the habitat of atlantids. High latitude ecosystems are a conveyor for OA, and organisms are already experiencing changes in their habitats. This, the environmental history, is crucial to understand and contextualize inter and intra-specific differences on the adaptive response to climate change

Line 135: We have added some oceanographic details here:

‘The SSTC is a narrow region at the boundary between warmer, more saline subtropical waters to the north, and the colder, fresher Sub-Antarctic Zone to the south [43]. As such, the SSTC is a highly variable region with strong gradients in salinity and temperature, and is expected to experience considerable ocean change, in particular OA [42,44,45].’

L129 Replace heteropods by atlantids. Transcriptome needs to be presented in the context of its contribution to solve the problem.

We feel it is important to highlight that this study is not only the first of its kind for atlantid heteropods, but also for any heteropod species. Although the three heteropod families are morphologically very distinct, the research presented here is relevant and valuable for all three families. For example, these genomic resources could prove to be very valuable for

phylogenomic and evolutionary studies of molluscs (e.g. Peijnenburg et al. 2020; Zapata et al. 2014).

L133-134. It would be illustrative to know the physical-chemical conditions shaping the environment of atlantids.

Line 135: Please see comment above. We have added oceanographic details for the region in which the study organism lives.

L128-142. I can't recognize a clear and testable hypothesis, neither the associated methods addressed in the current study. For example, for those questions stated between L133-137, how were these questions assessed? For clarity, this information needs to be provided to the readers.

Line 145: This is the first study of its kind and we do not have any priori hypotheses. Therefore we find it more appropriate to present research questions, rather than hypotheses. Within this section we have included three research questions and we have briefly described the methods used to address them 'We use a thorough multi-disciplinary approach, combining fluorescence microscopy (n=184) and micro-CT scanning (n=43) of the same individuals to quantify shell growth, as well as RNA sequencing of individuals from the same experiments to detect responses at the molecular level (n=6 samples of 8-10 pooled individuals) to address the following questions:...'. We feel it is more appropriate to describe the details of these methods within the methods section.

L146-152. In the Introduction section, authors focused shell functionality on adult stages. If the study was conducted with early development stages, information associated with the shell generation, linked or not to specific environmental regulation, should be mentioned in the Introduction section as well. Such as the authors mention in L152-154, these early development stages can show higher sensitivity to deviations in habitat suitability.

Line 79: We have added to the introduction (with citations):

'...despite likely being affected by imminent ocean changes, in particular rapid contemporary changes in ocean chemistry, and especially in the early life stages [9–12].'

L156-158. Whether juveniles are herbivorous, which function does the shell develop on these stages? Is it already present or is under development? Is there any relationship between shell production and environmental drivers? In other similar taxa? This information needs to be provided earlier to the readers.

Line 75: Very little is known about the function of the shell at the juvenile stage, but we know from personal observations that it is already important for protection – having often seen marks of failed predation even on very tiny shells. We have added to the introduction:

'The atlantid shell is already present upon hatching [3] and is likely important at the larval stage for physical protection, and also as ballast to allow rapid escape by sinking.'

L160-167. Given the information provided between L117-125, I was expecting the study was conducted at a high latitude ecosystem placed in the Atlantic basin.

Lines 112, 130, 139: As described in both Lines 180-185 and L129-L132, specimens for our study were collected in the Southern Subtropical Convergence Zone, a mid-high latitude ecosystem in the Atlantic. We have made this more clear in the text by referring to this as a 'mid-high latitude' region (e.g. Line 112), and describing the distribution of *Atlanta ariejansseni* as cold water (e.g. Lines 130, 139).

L175-178. To which depth this water belongs to (ambient water)?

Line 209: We have added the depth at which ambient water was collected:

‘Surface seawater (from ~10 m water depth) was filtered...’

L180. Which is the rationale of this treatment?

Line 195: We have moved this rationale of the treatments from the results/discussion section (also in response to Rev. 1). This section now begins:

‘To evaluate the effects of ocean carbonate chemistry on atlantid calcification, specimens of *A. ariejansseni* were incubated for three days across realistic recent-past, ambient and near-future pH scenarios. We applied a past scenario of 0.05 pH units higher than ambient (ambient pH 8.14 ± 0.02 , past pH 8.19 ± 0.02 , Table 1) that is approximately equivalent to the mid-1960s (assuming a decrease in pH of 0.001 units per year in this region) [18], and a future OA scenario of 0.11 pH units lower than ambient (pH 8.03 ± 0.00 , Table 1), which is approximately equivalent to expectations for the year 2050 in the South Atlantic Ocean (under IPCC Representative Concentration Pathway RCP8.5) [13,47].’

L182. Add the depth. Please see comment below.

L183. A relatively high and wide temperature range for a temperate ocean region. To which bathymetric range this temperature variation belongs to?

Line 216: The specimens were collected from 0–100 m water depth (oblique tows with an open net). We have added this information. The temperature was determined shortly after specimen sampling from the CTD profiles.

‘...at ambient ocean temperatures (at the depth of specimen collection 0–100 m)...’

L184. Mention if pH was measured with an electrode, spectrophotometer or probe, the precision, reference material, uncertainty (revise the Guide of good practices in OA studies).

Line 273: This information is included within the ‘Water chemistry’ methods section, and we have now added details of the reference material to that section.

‘pH was measured on the NBS scale using a research grade benchtop pH meter (HANNA HI5522-02, accuracy ± 0.002 , readable to 0.1 mV and 0.001 pH as recommended for OA research [39]) and a glass electrode. The pH meter was regularly calibrated using NBS standards (HANNA Millesimal Buffer Range, accuracy ± 0.002 pH).’

L189. Replace natural by wild.

Line 224: We have replaced ‘natural’ with ‘*in situ*’.

L189-194. Provide the water sampling depth for chlorophyll measurements and food supply.

Line 224: Thank you for the suggestion. We have now added the range of values, including depth (0-100 m).

‘*In situ* phytoplankton concentrations at the study sites over the depth sampled for specimens (0–100 m) was 0.16–0.42 $\mu\text{g/l}$ [48].’

L190-194. I don’t see the point of changing the wild food availability if authors are interested in characterizing responses to natural conditions. Whether current pH/pCO₂ ocean levels

constitute an environmental driver affecting phenotypic plasticity and genetic diversity of populations, ultimately selecting for adaptation to ocean acidification, the linkage between site-specific pH variability and physiological (for example) responses needs to be demonstrated without artificial perturbations that might hinder such adaptive process (Lewis et al., 2013; McElhany, 2017).

Lewis, C.N. et al., 2013. Proc. Natl. Acad. Sci., 110 (51), E4960–E4967.

McElhany, P. (2017). ICES Journal of Marine Science, 74(4), 926-928.

There are many limitations to laboratory studies. We have based our experimental feeding strategy on sound knowledge and experience. We have previously shown that atlantids calcify their shells close to the deep chlorophyll maximum (Wall-Palmer et al. 2018), so food is likely extremely important for calcification. It is also known that a lack of food has negative effects on the condition of shelled pteropods in experiments (the most comparable organisms to atlantids). For feeding under laboratory conditions, see Howes et al. 2014. Therefore we decided to provide a plentiful source of food to allow *ad libitum* feeding. We have explained this clearly within the manuscript. There have been some additions to this explanation in response to comments from Reviewers 1 and 2.

For example, Line 231:

A high concentration (~500-1000 times more than the *in situ* concentration) of algae was therefore added to ensure that *ad libitum* feeding was possible for the duration of the experiment.’

Howes EL *et al.* 2014 Sink and swim: a status review of thecosome pteropod culture techniques. Journal of Plankton Research 36, 299–315

Wall-Palmer et al. 2018. Vertical distribution and diurnal migration of atlantid heteropods, Marine Ecology Progress Series, 587, 1-15

L203. Were the food conditions and incubation water properties replaced during this period?
Line 229: The food and water were not replaced at any time during the experiments. We have made this more clear (also in response to Rev. 1) by including:

‘Algae was only added at the beginning of the experiment to prevent disturbance of the specimens and to avoid exposing the experimental seawater to the atmosphere.’

L211-218. This paragraph belongs to the Results section.

We have now separated the results (Line 397) and discussion (Line 552) sections.

L251-252. This paragraph belongs to the Results section.

We have now separated the results (Line 397) and discussion (Line 552) sections.

L254. Shell cleaning and fluorescent imaging to.... was carried out..

Line 287: We have changed the text as suggested and a definition of the shell extension has been added in response to Rev. 1.

‘One method for quantifying calcification is to measure the length of the piece of shell that grew during the experiment e.g. [1], herein described as the shell extension. Shell cleaning and fluorescent imaging to measure this shell extension was carried out...’

L286-287. This paragraph belongs to the Results section.

We have now separated the results (Line 397) and discussion (Line 552) sections.

L309-331. This paragraph belongs to the Methods section. Furthermore, the goals attained by utilizing this methodology and how its results help to test the hypothesis, need to be better and clearer presented within the Introduction section.

This section is already within the methods (now Lines 333-355). We have added a brief explanation of the RNA sequencing to the introduction, but we feel that a more detailed and thorough explanation is better suited to the methods section.

Line 147: ‘...as well as RNA sequencing of individuals from the same experiments to detect responses at the molecular level...’

L333-371. Same as above.

This section is already within the methods (now Lines 357-395).

L375. Rephrase. If the wild food environment was modified, these responses are not under ambient conditions.

Line 412: We have changed this to ‘Shell growth under ambient ocean chemistry conditions’.

L377-382. This paragraph mixed results and discussion and it needs to be edited.

We have now separated the results (Line 397) and discussion (Line 552) sections.

L384. “over three to eleven days” unclear sentence.

We have removed this from the sentence.

Line 424: ‘In the ambient growth experiment, specimens grew up to 99 μm (shell extension) per day (n=44, Table 2).’

L386-388. This paragraph mixed results and discussion and it needs to be edited.

We have now separated the results (Line 397) and discussion (Line 552) sections.

L393-395. This paragraph belongs to the Discussion section.

We have now separated the results (Line 397) and discussion (Line 552) sections.

L395-397. I couldn’t understand this sentence.

Line 434: We have made this more clear by changing the sentence to:

‘Adult specimens of *A. ariejansseni* were found to exhibit a comparatively low growth rate of $\sim 25 \mu\text{m}$ per day (mean of 5 adult specimens).’

L384-401. Most of the content of this paragraph has little incidence on the assessed treatments.

Line 424: This entire section refers to the separate growth rate experiment (described in the methods as ‘Growth rate experiment’) and not the OA experiments. This is now more clear because we separated the results (Line 397) and discussion (Line 552) sections.

L403-415. This complete section has no sense with regards to the form and content of the research article.

Line 656: This sentence reports an important finding of this study – a first estimation of the longevity of an atlantid species based on growth experiments. We have decided not to

remove it. We have included this in the discussion section (also in response to Reviewer 2).

L419-421. According to this rationale, pCO₂/pH variations select both ways the phenotypic plasticity and genetic diversity of populations. Therefore, calcification will increase as pH increases according to a certain pattern. It also assumes the distribution range both meridional and bathymetric of these populations before the industrial revolution, showed similar (and constant) pCO₂ values than the atmosphere. Thus, the role of the physical CO₂ pump operating and capturing atmospheric CO₂ at the Southern Ocean, since the establishment of the modern ocean circulation millions of years ago, and its evolutionary impact on species present even before, seems to be neglected.

Line 195: This section has now been moved to the methods section. As explained within this text, the three pH treatments are for the mid-1960s, ambient and 2050. We have not considered a time before the industrial revolution.

Discussion section needs to be synthesized and focused.

We have now separated the results (Line 397) and discussion (Line 552) sections. The discussion section is now much shorter and focussed.